# Disentangling five dimensions of animacy in human brain and behaviour

Kamila M. Jozwik [1✉], Elias Najarro [2], Jasper J. F. van den Bosch [3], Ian Charest [3], Radoslaw M. Cichy [4,6] & Nikolaus Kriegeskorte[5,6]

Distinguishing animate from inanimate things is of great behavioural importance. Despite distinct brain and behavioural responses to animate and inanimate things, it remains unclear which object properties drive these responses. Here, we investigate the importance of five object dimensions related to animacy ("being alive", "looking like an animal", "having agency", "having mobility", and "being unpredictable") in brain (fMRI, EEG) and behaviour (property and similarity judgements) of 19 participants. We used a stimulus set of 128 images, optimized by a genetic algorithm to disentangle these five dimensions. The five dimensions explained much variance in the similarity judgments. Each dimension explained significant variance in the brain representations (except, surprisingly, "being alive"), however, to a lesser extent than in behaviour. Different brain regions sensitive to animacy may represent distinct dimensions, either as accessible perceptual stepping stones toward detecting whether something is alive or because they are of behavioural importance in their own right.

[1] Department of Psychology, University of Cambridge, Cambridge, UK. [2] Digital Design Department, IT University of Copenhagen, Copenhagen, Denmark. [3] Département de Psychologie, Université de Montréal, Montreal, Canada. [4] Department of Education and Psychology, Freie Universität Berlin, Berlin, Germany. [5] Zuckerman Mind Brain Behavior Institute, Department of Psychology, Department of Neuroscience, Department of Electrical Engineering, Columbia University, New York, USA. [6] These authors jointly supervised this work: Radoslaw M. Cichy, Nikolaus Kriegeskorte. ✉email: jozwik.kamila@gmail.com

The perception of animate things is of great behavioural and evolutionary importance to humans and other animals. Recognizing animate things is essential for choosing appropriate actions as we engage in the physical and social world, and can be a matter of life and death (e.g., quick recognition of a predator). Animacy is an important representational division in nonhuman and human higher ventral visual cortical areas in the inferior temporal cortex[1] and the medial temporal lobe[2] as measured by functional magnetic resonance imaging (fMRI). Consistent with the importance of animacy perception in the classical neuropsychological literature, lesion studies established that living things are represented in dedicated regions of the cortex[3–5]. However, it is less clear which of the distinctive features of animate things are represented in the brain and reflected in judgments of animacy and of the similarity between things. An animal differs from an inanimate object in many respects, so animacy could be diagnosed by many different indicators. The dimensions of animacy that have been explored include "being alive"[6–12], "looking like an animal"[6,8,11–14], "having mobility"[15–17], "having agency"[17–23], and "being unpredictable"[20]. Each of these studies offers important insights on one particular dimension of animacy. However, to understand which of a number of confounded dimensions are represented, we need experiments designed to disentangle them.

Dimensions that define a concept depend on the chosen definition of the concept. Dictionary definitions of animate rely heavily on the dimension of being alive (animate: "living; having life", Oxford Advanced Learner's Dictionary). However, being alive is a difficult-to-assess latent property of a thing. It seems plausible that a perceptual system might represent more accessible dimensions that are correlated with being alive, even if being alive were ultimately the behaviourally important property. In addition, a more accessible related property, such as "looking like an animal" or "being unpredictable", may be of behavioural importance in its own right. We are not concerned here with the philosophical and semantic questions of animacy, but with the empirical question of which of several related and commonly conflated dimensions are represented in particular brain regions and in behavioural judgements.

Apart from the abovementioned dimensions of animacy, several other human-centred interpretations of animacy have been proposed. Recently reported animacy-related concepts that explain variance in the ventral visual stream fMRI measurements are human-likeness[24], humanness[23], resemblance to human faces and bodies[25], and capacity for self-movement and thought rather than face presence[26]. Another similar concept is the animacy continuum, where objects are perceived as more animate when they are more similar to humans (e.g., images of monkeys would be perceived as more animate than insects, even though both species belong to the animal category;[6]). In addition to the high-level dimensions, low-level visual features correlate with animacy[27–29]. Colour statistics[28], curvature[27], and mid-level shape features[29] can be used to classify whether an object is animate or inanimate.

The stimuli used in previous studies were mostly handpicked to investigate a chosen dimension without controlling for confounding variables. Previous studies often used unnatural stimuli (e.g., point-light displays in[17–22,30] or grey-scale simplified objects in[31]) or stimuli from only one category of objects (e.g., animals in[17–22,30]). The "being unpredictable" dimension has only been studied in the language domain and not in vision. Finally, previous studies mostly focused on one type of behaviour or one type of brain measurement but none of them has combined multiple measurements of behaviour and different brain measurement modalities, which would allow building a more comprehensive understanding. This approach has limited our progress toward a comprehensive understanding of the representation of the multiple dimensions of animacy. Thus, despite decades of research that has established the prominence of different dimensions of animacy in human brain and behaviour, it remains unclear whether any one of the five selected dimensions or a subset of them can explain the responses and how this depends on the brain region or behavioural measure.

Here, we comprehensively investigate the importance of the five selected dimensions of animacy: "being alive", "looking like an animal", "having agency", "having mobility", and "being unpredictable". By using a larger number and diversity of stimuli than in the previous studies, we can disentangle the dimensions experimentally. We study responses to this stimulus set using two behavioural tasks (animacy ratings and similarity judgements) and two brain measurement modalities (fMRI and EEG). To disentangle the five selected dimensions of animacy, we optimized a stimulus set of 128 images using a genetic algorithm (GA).

## Results

**Stimulus selection procedure and stimulus set.** Evaluating the contribution of individual dimensions of animacy ("being alive", "looking like an animal", "having mobility", "having agency", and "being unpredictable") would be best performed on a stimulus set that is as decorrelated on these dimensions as possible. We created such a stimulus set by optimizing stimuli using a genetic algorithm. We selected images that were maximally decorrelated on dimensions of animacy using a four-step procedure (Fig. 1a, see Methods for details). First, we created an animacy dimension grid where we asked participants to fill freely in the names of the objects fulfilling each animacy dimension combination to then find images that satisfy combinations of dimensions of animacy (29 out of 32 possible combinations). The object names provided by the subjects did not cover all 32 combinations, which is why 29 combinations out of 32 were included in step 1 of the stimulus selection procedure. Participants came up with 100 classes in total, and the participants were not given any object classes to use by the experimenters (Supplementary Table 1). The object categories were distinct (e.g., there were different types of robots and, therefore, two different object categories, "humanoid robot" and "animal robot", were included). Second, we assembled object images based on object names from step one. Third, an independent set of participants rated object images (which were assembled based on object names from step one) on each of the five selected dimensions of animacy to generate animacy ratings. Fourth, we used a genetic algorithm to select a subset of images with the lowest maximum correlation between dimensions of animacy (10,000 generations). The maximum correlation between dimensions in the stimulus set was 0.36. This result was better than when randomly selecting the stimuli 10,000 times without optimization (maximum correlation between dimensions = 0.64), proving that our novel stimulus selection procedure was successful. The pairwise correlations between animacy dimensions for the 128 stimuli selected randomly and for the 128 stimuli selected by GA are represented in Fig. 1b. This stimulus set was used in subsequent behavioural and brain imaging experiments.

The stimulus set (Fig. 2a) consisted of 128 images spanning almost all animacy dimension combinations (26 out of 32 possible combinations). Among 29 dimension combinations for which participants provided object names, three were not selected by the GA. This is because the objective of the GA was to minimise the maximum correlations between dimensions, and some dimension combinations were not optimal to be chosen. A wide range of objects was present, such as humans, human fetuses, human organs, human and animal shadows, plants, corals, forces of nature, game items, toys, vehicles, and electronic equipment covering 68 categories. The GA did not choose some

## a

**Step 1**. Create dimensions of animacy combination grid and acquire grid combination labels

| animacy dimensions | baby | robot | waves |
|---|---|---|---|
| being alive | 1 | 0 | 0 |
| looking like animal | 1 | 1 | 0 |
| having mobility | 1 | 1 | 1 |
| having agency | 1 | 0 | 0 |
| being unpredictable | 1 | 0 | 1 |

**Step 2**. Find 300 images corresponding to labels (3 images per label)

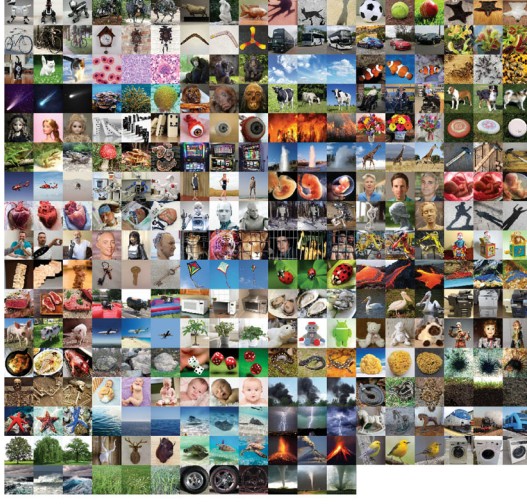

**Step 3**. Acquire ratings of animacy dimensions for 300 images

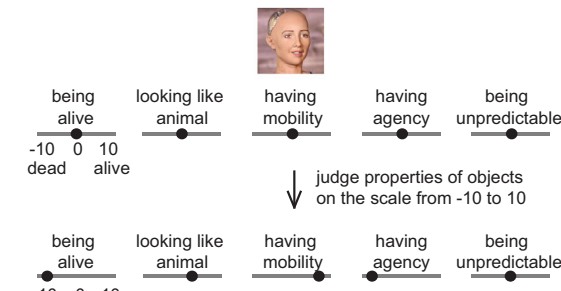

**Step 4**. Select 128 out of 300 images decorrelated on dimensions of animacy using genetic algorithm

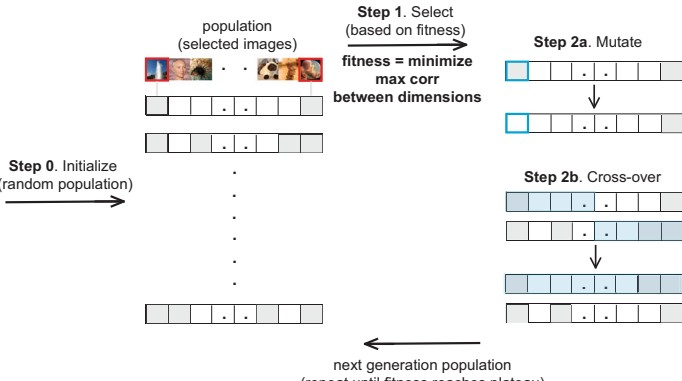

## b

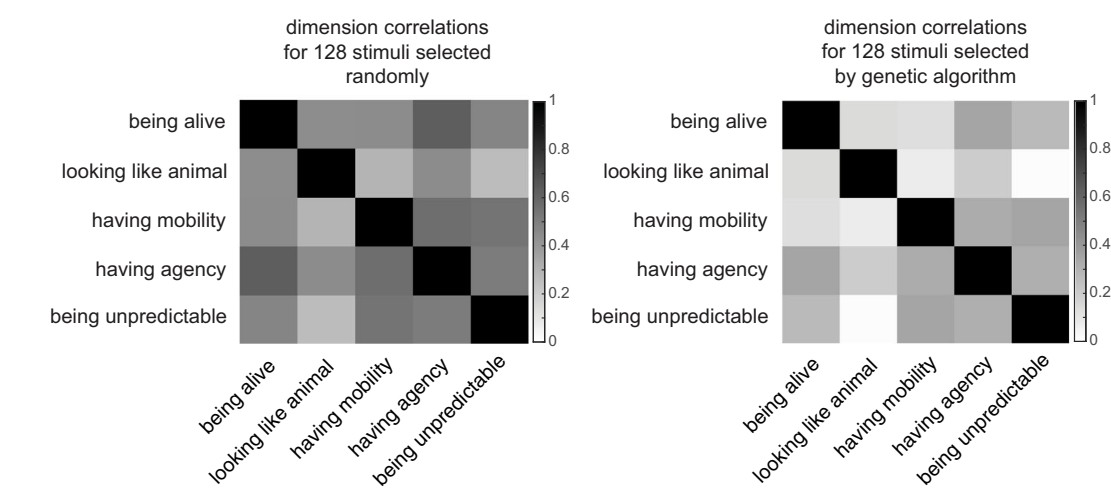

**Fig. 1 Stimulus-selection procedure and pairwise correlation between animacy dimensions before and after stimulus selection by the genetic algorithm. a** First, we created an animacy grid with all dimensions of animacy combinations and asked 11 participants to fill in the names of objects that fulfilled these combinations. Second, we assembled object images based on object names from step one. Third, an independent set of 26 participants performed animacy ratings of 300 of these object images. Finally, we selected an optimal set of stimuli that had a low correlation between dimensions (as behaviourally rated) using a genetic algorithm. These stimuli were used in behavioural and brain representation experiments where a new set of participants was recruited to make sure that the stimulus generation and the actual experiments were independent. **b** Pairwise correlation between animacy dimensions for the randomly selected 128 stimuli (left) and the 128 stimuli selected by the genetic algorithm (right) in behavioural ratings.

of the images representing object names from the initial 100 object names listed in step 1, as the dimensions of animacy ratings on these images were not optimal for the GA objective. The GA was allowed to choose a maximum of two different images representing different objects from a given category (e.g., two different animal robots) if this selection contributed to an optimal GA solution. A small percentage of stimuli can be considered as unusual (7.8%, 10 out of 128 stimuli, considering that stimuli such as human foetus, disembodied eyeball, person on life support, heart can be considered as emotional triggers). As the unusual stimuli constitute only a small percentage of our stimulus set we do not think that they would affect the

**a**

**b**

**Fig. 2 Stimulus set and study overview. a** The genetic-algorithm driven stimulus set consisted of 128 images decorrelated on dimensions of animacy. The stimuli were coloured images of sport equipment, games, robots, dolls and puppets, plush toys, land vehicles, air vehicles, plants, forces of nature (water, air, fire, smoke), sea organisms, cells, organs and fetuses, humans, food, kitchen and office equipment, shadows. **b** Study overview. All 19 participants performed two behavioural studies: animacy ratings and similarity judgements, and two brain response measurement studies: EEG (to access temporal information) and fMRI (to access spatial information). Importantly, participants first performed EEG and fMRI studies, then similarity judgements and finally animacy ratings. This experimental order was to ensure that participants did not know about animacy dimensions tested until the final animacy ratings.

interpretation of our results. None of the participants mentioned that they found any of the stimuli upsetting after performing the experiment. However, future studies may benefit from including affect ratings alongside the dimensions of animacy ratings. This stimulus set was used for two behavioural studies: animacy ratings and similarity judgements, and two brain response measurement studies: EEG (to access temporal information) and fMRI (to access spatial information). Nineteen participants performed all the studies. Importantly, participants first performed EEG and fMRI studies, followed by similarity judgements and finally animacy ratings (Fig. 2b). This experimental order was to ensure that participants did not know about animacy dimensions tested until the final animacy ratings.

**Consistency in animacy ratings across participants.** We first wanted to evaluate the contribution of each dimension of animacy when participants were asked to judge how animate an object image was. We first explored how consistent participants were in judging each dimension of animacy and each image. We used representational similarity analysis[32] to reveal which dimensions contribute most when participants judged animacy. We also examined which dimension(s) explained unique variance in the animacy ratings.

Participants judged each object image using a continuous scale from −10 to 10 for each dimension, e.g., −10 meant "inanimate"

and 10 meant "animate" (Fig. 3a). The same image was judged in the same way on the five investigated dimensions of animacy using the same scale. The mean between-participant correlation was 0.6, which indicated that participants were generally consistent in their ratings. The mean within-participant consistency for thirty repeated stimuli was 0.89 meaning that participants consistently judged the same stimulus within a session. The raw data of animacy ratings shows that a given stimulus could be differently rated on each of the dimensions (Fig. 3b).

We wanted to know how consistent participants were in judging each dimension and each stimulus. This analysis had two purposes: to be able to determine whether the variability in animacy ratings is low enough to interpret the ratings at all and to test which dimensions and stimuli were particularly controversial for participants as reflected by higher variability.

We first asked about the participants' consistency across stimuli in all dimensions of animacy. There is variability in the consistency of ratings as some of the object images, e.g., "bike", was judged very consistently as expected, whereas other more controversial ones, e.g., 'human foetus', less consistently, with "human shadow" having the lowest consistency among the object images tested (Fig. 3c). While participants were asked to judge what is represented on images ("human shadow"), it might not be apparent whether to judge the shadow itself or the person creating the shadow.

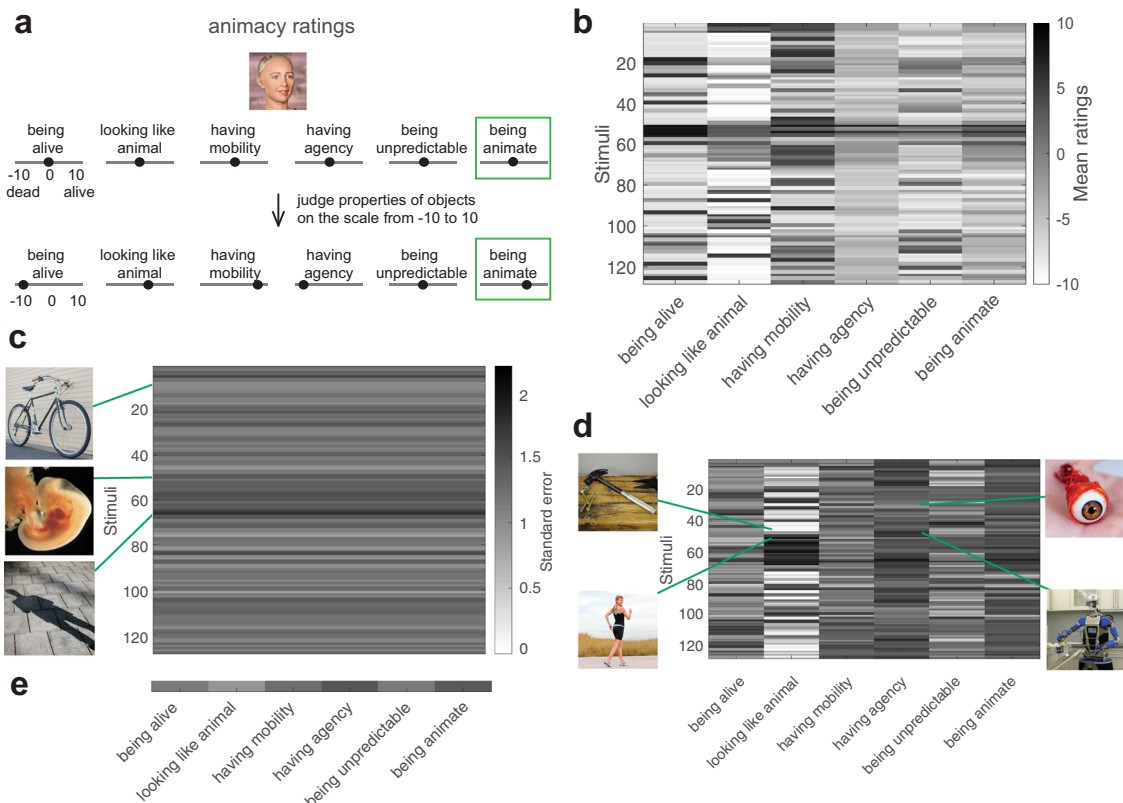

**Fig. 3 Animacy ratings and their consistency with examples of images judged consistently and not very consistently. a** Illustration of animacy ratings. Participants judged each object image using a continuous scale from −10 to +10 for each animacy dimension, e.g., −10 meant "dead" and +10 meant "alive" for the "being alive" dimension. Additionally, participants performed a rating of "being animate" dimension in a similar fashion. **b** Mean ratings of each animacy dimension and stimulus across 19 participants. **c** Consistency of each stimulus in animacy ratings across participants (standard error of the mean) with examples of stimuli with varying values of standard error. **d** Consistency of each animacy dimension and stimulus in animacy ratings across participants with examples of stimuli with varying values of standard error for the most consistently judged ("looking like an animal") and the least consistently judged ("having agency") dimensions. **e** Consistency of each animacy dimension in animacy ratings across participants (standard error of the mean).

To get more insights into animacy dimension ratings, we explored the consistency of each stimulus per each animacy dimension. For "looking like an animal", which was judged most consistently among the dimensions, "hammer" was one of the object images that were most consistently judged, whereas an image of a "human" was not judged very consistently. The lower consistency of judging an image of a human may be related to the fact that some humans do and others do not consider themselves animals, even though from a biological point of view Homo sapiens belong to the animal category. Looking at the other side of the spectrum - "having agency" dimension was judged least consistently - we observed that an example image that had a high consistency of ratings was "eyeball", whereas an image of a "robot" was not very consistently judged (Fig. 3d).

Among all the dimensions participants judged "having agency" least consistently and "looking like an animal" most consistently (see Fig. 3e for the consistency of ratings for each dimension). What about the animacy ratings ("being animate")? Would object animacy be judged consistently across participants or given the ambiguous definition of this term, would the consistency be lower than some of the more precisely defined dimensions of animacy? We found that the latter was the case - animacy ratings had lower consistency than more precisely defined dimensions of animacy, except for the 'having agency' dimension.

**Contribution of each animacy dimension to animacy ratings.** To gain more intuition about what stimuli are considered to have

the highest value on each animacy dimension, we visualized 10 object images with the overall minimum and 10 with the maximum rating on a given dimension (Fig. 4a). Overall, images that had low values on animacy dimension ratings were similar among dimensions (e.g., plush toys, meat, washing machine). In contrast, images with high ratings did differ depending on a dimension tested (e.g., stimuli judged as the most unpredictable being humans and forces of nature, in contrast to humans judged as having the most agency), proving that indeed these dimensions capture different aspects of animacy perception (Fig. 4a, right panel).

To reveal which dimensions contribute most when participants judged animacy we used representational similarity analysis (RSA, see Methods). RSA characterises representations in behavioural and brain data and models by representational dissimilarity matrices (RDMs) of the response patterns elicited by stimuli. RDMs capture the information represented by data or models by characterizing their representational geometry[33,34]. The representational geometry reflects which stimulus information is emphasized and which is de-emphasized. Models (here dimensions of animacy) are tested by correlating their RDMs with data RDMs.

Using RSA, we found that among all the dimensions, "having agency" and "being alive" explained more variance than some other dimensions (Fig. 4b) in animacy ratings (when participants were asked to judge how animate an object image was). This result means that when asked to judge animacy humans mostly think about whether an object is alive and whether it has agency. Even though "having agency" and "being alive" explain more variance than the other dimensions, it does

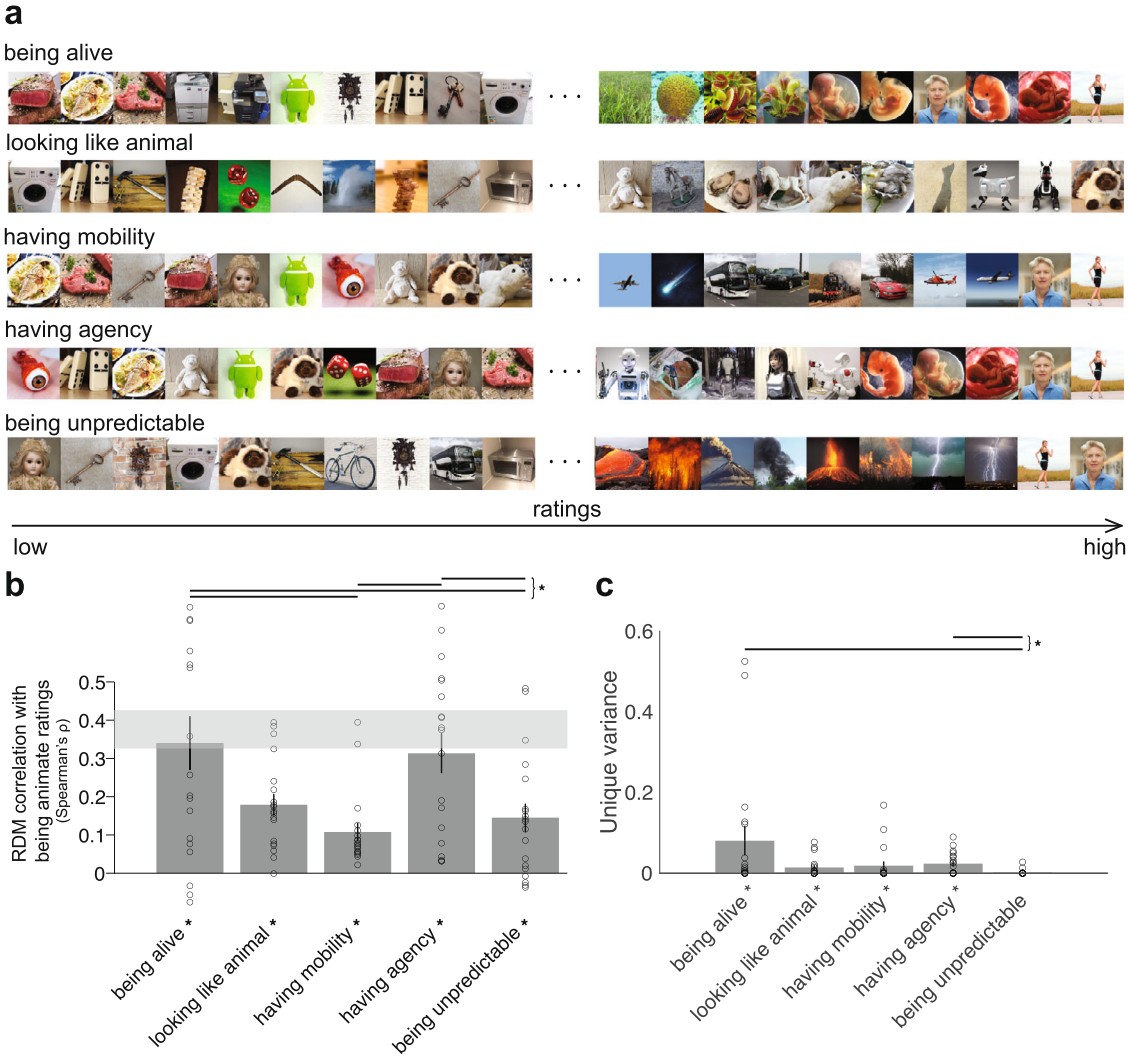

**Fig. 4 Dimensions of animacy and animacy ratings. a** Order of images with lowest and highest ratings on each animacy dimension. Out of 128 images, we show ten lowest and ten highest rated images on each animacy dimension. **b** Animacy dimension representational dissimilarity matrices (RDMs) comparisons with animacy ratings ("being animate") RDMs. Bars show the correlation between the animacy ratings RDMs and each animacy dimension RDM of 19 participants. A significant correlation is indicated by an asterisk (one-sided Wilcoxon signed-rank test, $p < 0.05$ corrected). Error bars show the standard error of the mean based on single-participant correlations, i.e., correlations between the single-participant animacy ratings RDMs and animacy dimension RDM. Circles show single-participant correlations. The grey bar represents the noise ceiling, which indicates the expected performance of the true model given the noise in the data. Horizontal lines show significant pairwise differences between model (here dimensions of animacy) performance ($p < 0.05$, FDR corrected across all comparisons), an asterisk to the right of horizontal lines indicates their significance. **c** Unique variance of each animacy dimension in explaining animacy ratings computed using a general linear model (GLM). For each animacy dimension m, the unique variance was computed by subtracting the total variance explained by the reduced GLM (excluding the dimension of interest) from the total variance explained by the full GLM. Specifically, for dimension m, we fit GLM on X = "all dimensions but m" and Y = data, then we subtract the resulting $R2$ from the total $R2$ (fit GLM on X = "all dimensions" and Y = data). We used non-negative least squares to find optimal weights. A significant unique variance is indicated by an asterisk (one-sided Wilcoxon signed-rank test, $p < 0.05$ corrected). The error bars show the standard error of the mean based on single-participant unique variance. Circles show single-participant unique variance. Horizontal lines show significant pairwise differences between model performance ($p < 0.05$, FDR corrected across all comparisons), an asterisk to the right of horizontal lines indicates their significance.

not mean that they explain unique variance. To test that, we performed a unique variance analysis (see Methods for details) and observed that, "being alive" and "having agency" explain significantly more unique variance than "being unpredictable" dimension (Fig. 4c). "Being alive" and "having agency" are the dimensions that explain the most variance in animacy ratings and also the ones that explain significantly more unique variance than some of the other dimensions.

In summary, when asked to rate object animacy, participants found "being alive" and "having agency" as dominant dimensions, and were most consistent when judging the "looking like an animal" dimension and least consistent in judging the "having agency" dimension.

**Contribution of each animacy dimension to similarity judgements.** After having established the contribution of animacy dimensions to animacy ratings, we tested whether any of the dimensions would explain similarity judgements. The similarity judgements task allows participants to evaluate objects in a more natural way concentrating on general object similarity. Please note that participants performed the similarity judgements task before competing dimensions of animacy ratings, so they were

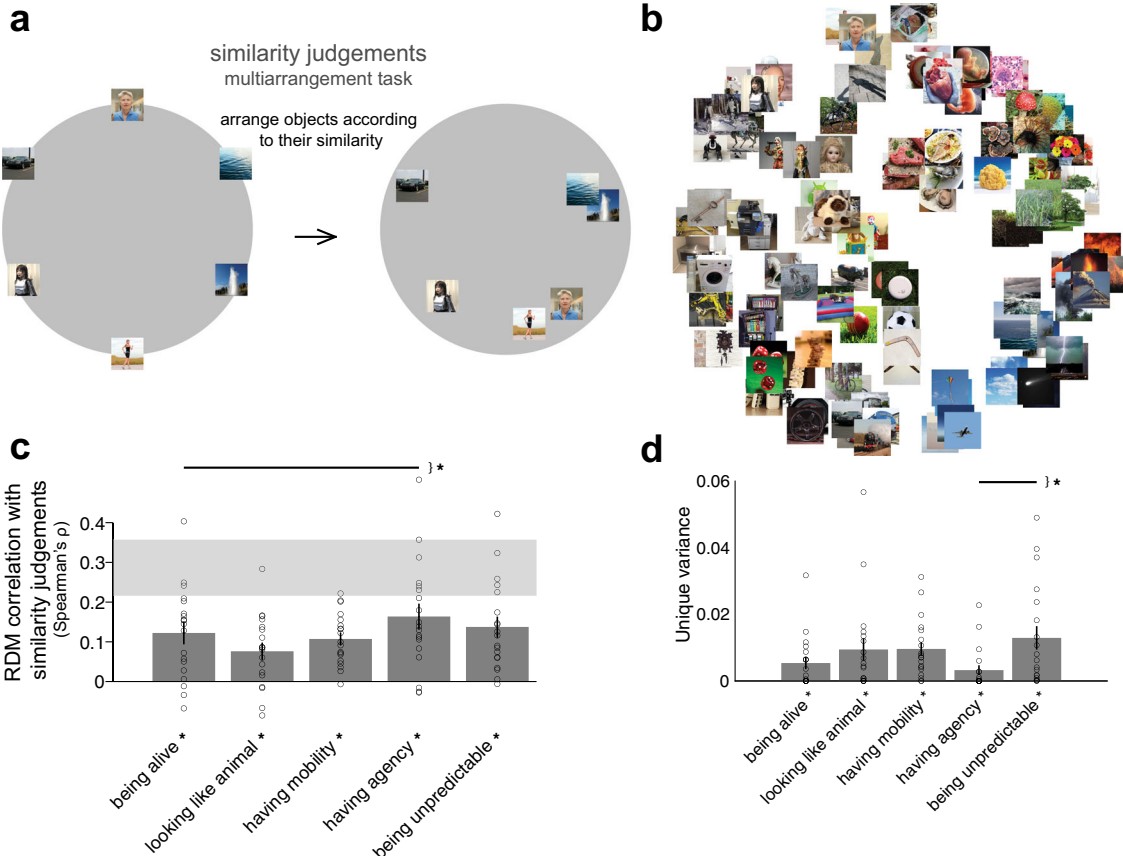

**Fig. 5 Dimensions of animacy and similarity judgements. a** Similarity judgements multiarrangement task. During this task, object images were shown on a computer screen in a circular arena, and participants were asked to arrange the objects according to their similarity, such that similar objects were placed close together and dissimilar objects were placed further apart. Participants performed multiple arrangements of subsets of the images, enabling us to estimate the underlying perceptual similarity space (see Methods for details). **b** Multidimensional scaling plot of similarity judgements (mean across 19 participants, with metric stress criterion). **c** Animacy dimension RDM comparisons with similarity judgements RDMs. Bars show the correlation between the similarity judgements RDMs and each animacy dimension RDM. A significant correlation is indicated by an asterisk (one-sided Wilcoxon signed-rank test, $p < 0.05$ corrected). Error bars show the standard error of the mean based on single-participant correlations, i.e., correlations between the single-participant similarity judgements RDMs and animacy dimension RDM. Circles show single-participant correlations. The grey bar represents the noise ceiling, which indicates the expected performance of the true model given the noise in the data. Horizontal lines show significant pairwise differences between model performance ($p < 0.05$, FDR corrected across all comparisons), an asterisk to the right of horizontal lines indicates their significance. **d** Unique variance of each animacy dimension in explaining similarity judgements. For each animacy dimension m, the unique variance was computed by subtracting the total variance explained by the reduced GLM (excluding the dimension of interest) from the total variance explained by the full GLM. Specifically, for dimension m, we fit GLM on X = "all dimensions but m" and Y = data, then we subtract the resulting R2 from the total R2 (fit GLM on X = "all dimensions" and Y = data). We used non-negative least squares to find optimal weights. A significant unique variance is indicated by an asterisk (one-sided Wilcoxon signed-rank test, $p < 0.05$ corrected). The error bars show the standard error of the mean based on single-participant unique variance. Circles show single-participant unique variance. Horizontal lines show significant pairwise differences between model performance ($p < 0.05$, FDR corrected across all comparisons), an asterisk to the right of horizontal lines indicates their significance.

unaware of the dimensions of animacy tested. As participants were asked to arrange object images based on their similarity and not asked about animacy, dimensions of animacy may not explain these representations. Rather than dimensions of animacy, either other categorical divisions or lower-level image features could be used for judging object similarity. Participants placed images of objects inside a circular arena according to how similar they judge them (Fig. 5a). The procedure was repeated with different numbers of objects that had to be arranged indefinitely until reaching a predefined arrangement consistency (see Methods). We evaluated how well the dimensions of animacy explained the similarity judgements task using RSA and the unique variance analysis.

Visualizing the similarity judgements as a multidimensional scaling (MDS) plot helped us to determine which object images were grouped together (Fig. 5b). For example, object images of

most robots, fetuses, and a human on life support were grouped, with human images separated but placed close to an image of a realistic humanoid robot. This grouping is related to the agency dimension as these images received the highest rating on "having agency" dimension (Fig. 5b). Animal robots formed their own cluster together with other moving objects such as boomerangs, balls, and buses. Different vehicles were grouped nearby with images of cars being placed near comets, clouds, and dominos. Games and toys were arranged in proximity to the "animal-like robot" group and the "human" group. Some unpredictable objects were also grouped together: geysers and game machines, or volcanos and waves (Fig. 5b). As a sanity check, images that depict the same object were grouped together, for example, two pictures of flowers or wheels (Fig. 5b). To gain an intuition of how well the five dimensions of animacy studied here separated representations in the similarity judgements, we have displayed

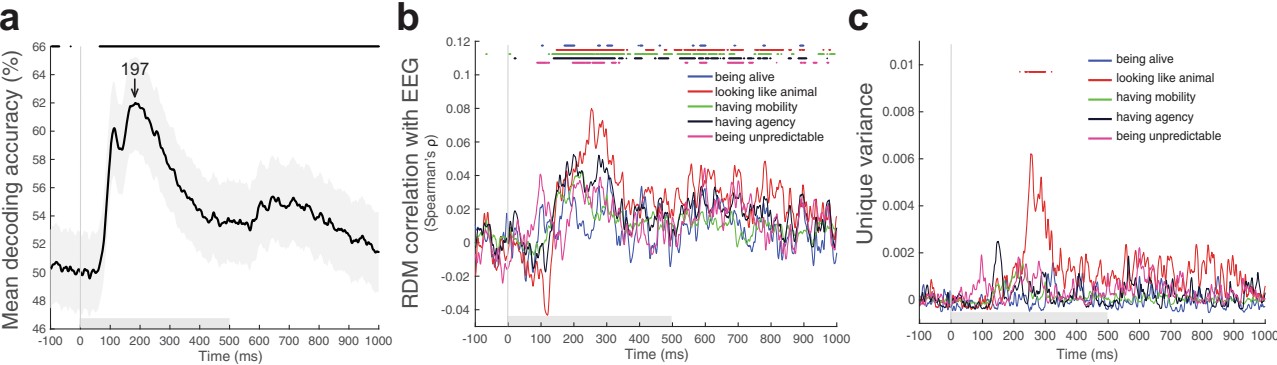

**Fig. 6 Dimensions of animacy and EEG time course. a** Mean decoding curve across 19 participants (pairwise stimuli decoding using a support vector machine approach). Significant decoding is indicated by a horizontal line above the graph (one-sided Wilcoxon signed-rank test, $p < 0.05$ corrected) and starts at 43 ms ($+/−2$ ms, standard error) with a peak latency of 197 ms ($+/−7$ ms, standard error, indicated by an arrow). The shaded area around the lines shows the standard error of the mean based on single-participant decoding. The grey horizontal bar on the x axis indicates the stimulus duration. **b** Animacy dimension RDM comparison with EEG RDMs across time. Lines show the correlation between the EEG RDMs and each animacy dimension RDM. A significant correlation is indicated by a horizontal line above the graph (one-sided Wilcoxon signed-rank test, $p < 0.05$ corrected). The grey horizontal bar on the x-axis indicates the stimulus duration. **c** Unique variance of each animacy dimension in explaining EEG RDMs computed using a GLM. For each animacy dimension, the unique variance is computed by subtracting the total variance explained by the reduced GLM (excluding the animacy dimension of interest) from the total variance explained by the full GLM, using non-negative least squares to find optimal weights. A significant unique variance (between 237 and 301 ms) is indicated by a horizontal line above the graph (one-sided Wilcoxon signed-rank test, $p < 0.05$ corrected). The grey horizontal bar on the x axis indicates the stimulus duration.

MDS plots colour-coded according to binary animacy dimensions (e.g., "being alive" with one colour of dots and "not being alive" with another colour of dots, Supplementary Fig. 1). All dimensions separated the stimuli well, with each dimension revealing different divisions between stimuli (Supplementary Fig. 1). Despite "having agency" dimension having positive values for three stimuli (images of two adult humans and a human foetus in the late stages of pregnancy, Supplementary Fig. 1), this dimension of animacy explained significant variance in the similarity judgements (Fig. 5c).

If we assume that the similarity judgements are based only on the similarity between low-level visual features, the dimensions of animacy should not explain a large fraction of the variance. This assumption is not what we observed - all dimensions of animacy explained a significant amount of variance in the similarity judgements task (Fig. 5c). If the low-level features were the only ones that participants used in similarity judgements object arrangements, then we would see on the MDS that objects are arranged by, for example, colour or shape, but this is not what we observe (Fig. 5b). None of the dimensions fully explained the similarity judgements data but the "having agency" dimension was close to explaining the total explainable variance given the noise in the data. As a portion of the variance remained unexplained, other dimensions beyond the ones explored here are likely needed to capture the data fully. Overall, when judging object similarity, humans use all dimensions of animacy tested here.

Even when all dimensions of animacy explain similarity judgements, maybe one or more dimensions explain unique variance. After having performed the unique variance analysis, we observed that each dimension explained unique variance in the similarity judgements (Fig. 5d). However, almost no dimension explained more unique variance than the other dimensions. This finding suggests that each dimension not only explains variance in the data but also explains a unique portion of that variance.

It is important to consider the effect of low-level visual features on the interpretation of these results and test whether the five animacy dimensions studied here explain unique variance over and above low-level visual features in similarity judgements. We included the first convolutional layer of the deep neural network

AlexNet[35] as a model of low-level visual features in this analysis. We observed that each of the animacy dimensions explained a significant amount of unique variance over and above the variance explained by low-level visual features and the other dimensions (Supplementary Fig. 2).

Overall, each animacy dimension explained a similar amount of variance in a behavioural task of similarity judgements, meaning that participants use higher-level dimensions of animacy when judging object similarity.

**Contribution of each animacy dimension to EEG time course.** Having shown the contribution of each animacy dimension in explaining animacy ratings and in a behavioural task of similarity judgements, we asked whether dimensions of animacy explain the time course of object image processing in the brain using EEG. One possibility is that once an image is shown for only half a second, the brain performs only automated image processing, and higher-level dimensions of animacy do not contribute to this process at all. The other possibility is that beyond low-level features, higher-level dimensions of animacy do play a role in forming brain representations of images even with a short stimulus presentation. To arbitrate between those possibilities, we evaluated the amount of variance explained by each animacy dimension and tested whether any dimension(s) explains unique variance in the EEG signal.

We first performed multivariate pattern analysis (MVPA) to determine how well we could decode the pattern of activations evoked by each stimulus. We performed pairwise stimuli decoding using a support vector machine approach and we could decode images in the stimulus set to a high decoding accuracy (62%) in a long time window (between 43 and 1000 ms after stimulus onset, Fig. 6a). The peak decoding accuracy was at 197 ms ($+/- 7$ ms, standard error).

To explore the structure of the representations, we displayed MDS plots for the selected timepoints: 0 ms - when the stimulus was just displayed, 100 ms - when the decoding accuracy started to go up, 200 ms - peak decoding accuracy, and 300 ms - when the decoding accuracy started to drop. As expected, no structure was visible at the stimulus onset (0 ms, Supplementary Fig. 5a). At

100 ms, human and humanoid and animal robot faces were grouped together, as well as forces of nature (Supplementary Fig. 5b). At 200 ms, faces were still grouped together, however, faces of a human and robots were represented further away from each other (Supplementary Fig. 5c). We have also displayed MDS plots colour-coded according to binary animacy dimensions for visualization purposes (Supplementary Fig. 6). At 300 ms, we observed similar clusters to those present at 100 ms (Supplementary Fig. 5d).

Once we knew that we could decode our stimuli, we asked how much variance each animacy dimension explained in EEG recordings. Do any of the dimensions explain any variance at all? To answer this question, we correlated each animacy dimension with EEG representations at every time point in every participant using RSA. First, we determined whether each of the animacy dimension RDMs was significantly related to the EEG data RDMs at every timepoint using a participant-as-random-effect analysis (one-sided Wilcoxon signed-rank test). We subsequently tested for differences in animacy dimension performance between each pair of dimensions of animacy at each timepoint using a participant-as-random-effect analysis (two-sided Wilcoxon signed-rank test). We accounted for multiple comparisons for each analysis by controlling the FDR at 0.05. We found that most animacy dimensions explained a significant amount of variance in EEG recordings (Fig. 6b, Supplementary Fig. 3); however, some dimensions explained variance at slightly different times. Despite differences in the exact timing of when dimensions of animacy explained the variance, a very clear pattern that one dimension explains representations earlier than the other was not observed. However, 'being unpredictable' explained significantly more variance than most dimensions in early time points: specifically more than "looking like an animal" (89–130 ms), "having mobility" (89–113 ms), and "having agency" (79–126 ms). While "looking like an animal" explained more variance than most other dimensions in later time points: more than "being alive" (209–302 ms), "having agency" (230–266 ms), and "being unpredictable" (146–184 ms). Finally, "having agency" explained more variance than most of the dimensions even later in time: more than "being alive" (268–301 ms), "having mobility" (261–289 ms) and "being unpredictable" (293–315 ms). To investigate how the five dimensions of animacy studied here relate to the general animacy, we correlated the general animacy ratings alongside the dimensions of animacy with the EEG RDMs. We observed that the "being animate" ratings explained a significant amount of variance in the EEG responses, similar in magnitude and timing to the variance explained by the dimensions of animacy tested (Supplementary Fig. 4).

Most animacy dimensions explained variance in EEG recordings. Is it the same or unique variance? Does one dimension explain more unique variance than the others, as in the case of animacy ratings, or is there no difference between the amount of unique variance explained by each dimension, as for the similarity judgements? We found that only one dimension -"looking like an animal"- explained the unique variance in EEG recordings between 237 and 301 ms. None of the other dimensions explained any significant unique variance (Fig. 6c). We wanted to check whether the unique variance explained by the "looking like an animal" dimension in the EEG data can be related to low-level visual features or whether this dimension explains unique variance over and above the variance explained by low-level features. We therefore included the first convolutional layer of AlexNet as a model of low-level visual features in the unique variance analysis in addition to the five dimensions of animacy studied here. We observed that our result of "looking like an animal" explaining a significant amount of the unique variance

still holds. "Looking like an animal" explains the unique variance not explained by either the other four dimensions or the first convolutional layer of AlexNet (Supplementary Fig. 7).

Overall, most dimensions of animacy explained EEG recordings with subtle differences in timing, but only 'looking like an animal' explained unique variance. Even for the rapid object recognition time course, higher-level dimensions of animacy explained a significant amount of variance in brain representations.

**Contribution of each animacy dimension to fMRI representations.** We asked where in the brain dimensions of animacy explain patterns of responses to images using fMRI. We performed both regions of interest (ROI) analysis along the ventral and dorsal visual streams and searchlight analysis. The ROI analysis was performed to evaluate the contribution of dimensions of animacy in the brain regions along the visual stream where we know object images are represented. The searchlight analysis complemented the ROI analysis testing whether other regions in the visual stream exist where dimensions of animacy explain variance that we may have missed when preselecting ROIs.

We first evaluated the contribution of each animacy dimension in ROIs across the ventral: visual area 1 (V1v), ventral occipital cortex 2 (VO2), parahippocampal cortex 2 (PHC2) and dorsal: visual area 1 (V1d), lateral occipital cortex 2 (LO2), TO2 visual streams (Fig. 7a, Supplementary Fig. 9 - with displayed noise ceiling). To define ROIs, we used a Probabilistic brain atlas (Wang et al., 2015). The locations of the examined ROIs are presented in Supplementary Fig. 8. In the ventral visual stream, "being unpredictable", "having mobility", and to a lesser extent "having agency" explained variance in V1v, in contrast to higher-level visual areas (VO2, PHC2) where additionally "looking like an animal" explained a significant amount of variance and "having agency" explained more variance than in V1v. In the dorsal visual stream, "being unpredictable" and "having mobility" explained variance in V1d, with "having agency" explaining variance in higher-level visual areas (LO2, TO2), and "looking like an animal" explaining variance only in TO2. As the dorsal stream carries information related to movement it is intuitive that "having mobility" and "being unpredictable" explain variance in dorsal regions. It was not clear, however, whether "having agency" and "looking like an animal" would explain variance in dorsal regions but it is indeed what we observe. After examining the contribution of general "being animate" ratings to the fMRI ROI representations, we did not see that "being animate" explained a significant amount of variance in fMRI data (except PHC2, Supplementary Fig. 10). For completeness, we performed the same analysis for all regions in ventral (V1v, V2v, hV4, VO1, VO2, PHC1, PHC2) and dorsal (V1d, V2d, V3d, V3a, V3b, LO1, LO2, TO1, TO2) visual streams. We observed that more dimensions of animacy studied here explain variance in higher-level visual cortex than in early visual cortex (Supplementary Figs. 11 and 12).

To gain an intuition about the structure of the representations, we displayed MDS plots for the ROIs. As expected, the stimuli in early visual regions were grouped by shapes and colours, whereas we could see clusters of faces and forces of nature in higher-level visual regions (Supplementary Fig. 13). In addition, the colour-coded MDS plot for PHC2 based on the binary dimensions of animacy revealed mild categorical structure for some dimensions (e.g., "looking like an animal", Supplementary Fig. 14).

The results of the unique variance analysis are included in Supplementary Fig. 15. "Being unpredictable", "having mobility", and "having agency" dimensions explained unique variance in early visual cortex. Except "being alive" dimension, four dimensions of animacy explained unique variance in higher-

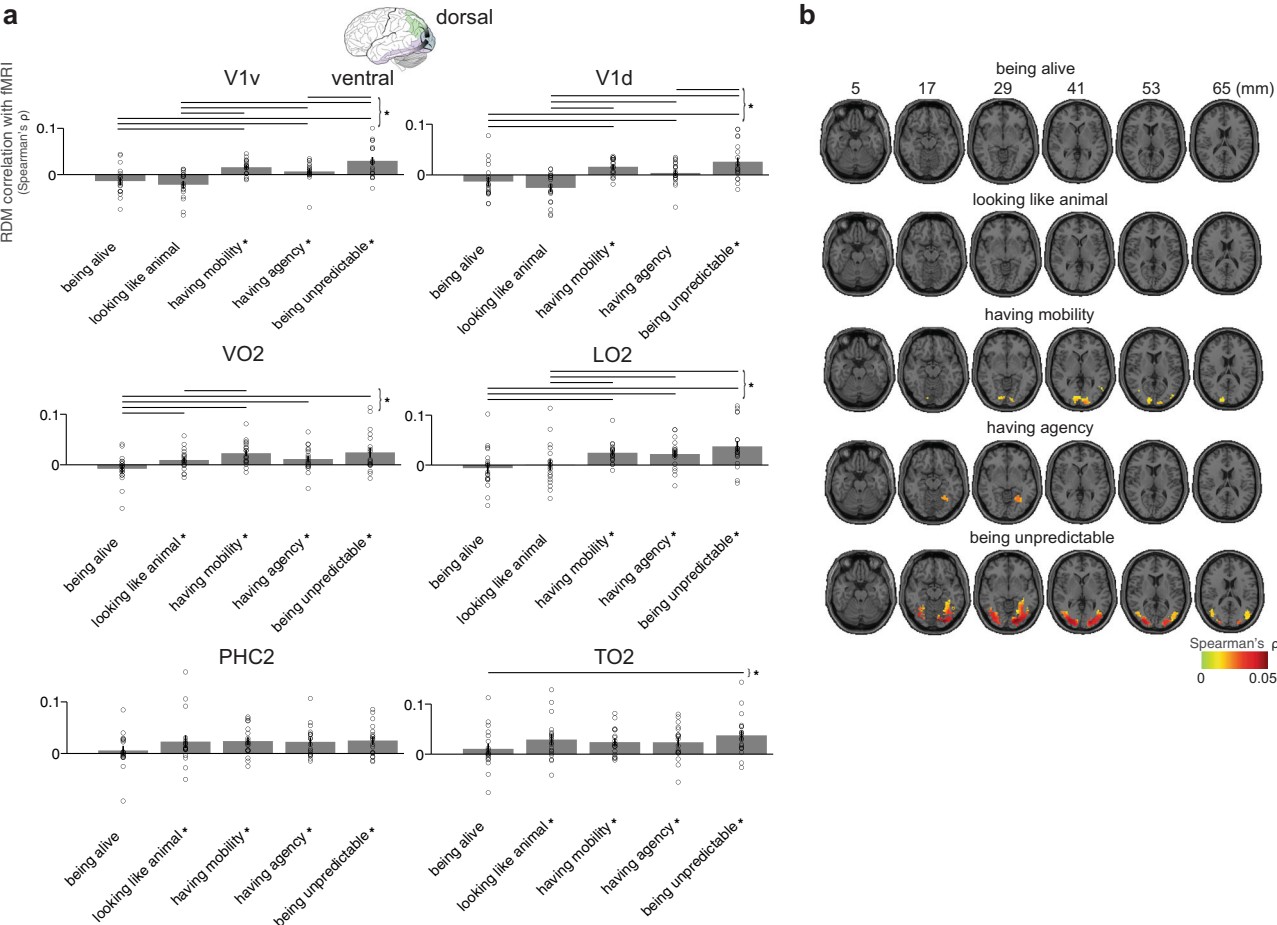

**Fig. 7 Dimensions of animacy and fMRI responses. a** Animacy dimension RDM comparisons with fMRI ROI RDMs of 19 participants. Bars show the correlation between each animacy dimension RDM with fMRI ROI RDMs. We selected ROIs across the ventral (V1v, VO2, PHC2) and dorsal (V1d, LO2, TO2) visual streams. A significant correlation is indicated by an asterisk (one-sided Wilcoxon signed-rank test, $p < 0.05$ corrected). Error bars show the standard error of the mean based on single-participant correlations, i.e., correlations between the single-participant ROI RDMs and animacy dimension RDM. Circles show single-participant correlations. Horizontal lines show significant pairwise differences between model performance ($p < 0.05$, FDR corrected across all comparisons), an asterisk to the right of horizontal lines indicates their significance. **b** Searchlight analysis with each animacy dimension showing where in the brain animacy dimensions explain image representations masked with the visual stream regions (Spearman's ρ between animacy dimension and brain representations, one-sided Wilcoxon signed-rank test, FDR controlled at 0.05).

level visual areas. We also wanted to test whether the five dimensions of animacy studied here explained variance over and above low-level visual features in the fMRI ROI analysis. We therefore included the first convolutional layer of AlexNet along with the five dimensions of animacy in the unique variance analysis. Similarly to behavioural and EEG results, selected dimensions of animacy explained variance over and above low-level visual features of AlexNet's first convolutional layer (Supplementary Fig. 16). As expected, the unique contribution of the first convolutional layer of AlexNet was greater in the early and intermediate visual cortex in comparison to the high-level visual cortex (Supplementary Fig. 16). These results suggest that the dimensions of animacy are important for both ventral and dorsal visual streams, however, a substantial variance remained unexplained.

To investigate the contribution of each animacy dimension in a spatially unbiased fashion we performed a searchlight analysis in the visual stream (Fig. 7b). Consistently with the ROI analysis "being alive" did not explain any significant amount of variance in the brain. "Looking like an animal" also did not explain any significant amount of variance in searchlight analysis but it is possible that with a larger amount of data we would see this dimension explaining some variance as the ROI analysis and EEG

results pointed in that direction. "Having mobility" dimension explained variance in early visual cortex only (based on the correlation strength of the searchlight analysis). In contrast, "having agency" explained variance only in higher-level visual cortex, which is consistent with the ROI analysis where "having agency" explains more variance in higher-level visual areas and further suggests that this dimension best captures higher-level representations. One dimension that explained variance in both early and higher-level visual areas was "being unpredictable" suggesting that unpredictability is important for attention and is already detected in the early visual cortex. Despite the living non-living distinction being thought to be important for brain representations "being alive" dimension did not explain any significant amount of variance in brain responses based on the ROI and searchlight analyses. This suggests that the ventral and dorsal visual streams do not represent the results of a deeper cognitive inference process that would assess whether something is alive.

## Discussion
We investigated the representation of the five selected facets of animacy in brain and behaviour (summary in Table 1) and

**Table 1 Summary of findings.**

| | |
|---|---|
| **being alive** | explains no significant variance for brain representations in this study (but strongly reflected in behavioural judgments like the other four dimensions) |
| **looking like animal** | only dimension explaining significant unique variance in EEG (weak significant effects in fMRI) |
| **having agency** | explains significant variance in higher-, but not lower-level visual cortex in fMRI (weak significant effects also in EEG) |
| **having mobility & being unpredictable** | both dimensions explain small amounts of significant variance in lower- and higher-level visual cortex in fMRI (for both, there are weak significant effects also in EEG) |

We find that the five tested dimensions of animacy captured behaviour very well. Brain representations were also explained by most dimensions (surprisingly not "being alive"), however, to a lesser extent than behaviour. The living/non-living distinction ("being alive") features prominently in both dictionary definitions of animacy and the neuroscience literature on brain representations. Consistent with this prominent role, "being alive" accounted for about half the explainable variance in our participants' object similarity judgements. Surprisingly, however, "being alive" did not explain variance in brain representations. The other four dimensions of animacy explained variance in both brain and behaviour. The "looking like an animal" dimension was the only dimension that explained significant unique variance in the EEG data. One interpretation is that "looking like an animal" provides an accessible visual correlate of animacy that can be computed by the visual system. The "having agency" dimension explained more variance in higher-level visual areas, consistent with the cognitive demands of determining agency. The "being unpredictable" dimension was reflected in representations in both lower and higher-level visual cortex, possibly because unpredictable things require attention. In the fMRI data, three dimensions explained unique variance in early visual cortex ("having agency", "having mobility", and "being unpredictable") and four dimensions in higher visual cortex (all except "being alive"). Our results reveal that different brain regions sensitive to animacy may actually represent distinct dimensions, either as accessible perceptual stepping stones toward detecting whether something is alive or because they are of behavioural importance in their own right.

concluded that different brain regions sensitive to animacy may actually represent distinct dimensions, either as accessible perceptual stepping stones toward detecting whether something is alive or because they are of behavioural importance in their own right. Including multiple dimensions in a linear model enabled us to disentangle their roles. To increase the power of these analyses, we decorrelated the five selected dimensions of animacy using an optimized set of natural stimuli. We managed to reduce the maximum pairwise correlation from 0.64 (for random selection) to 0.36 (optimized). It may not be possible to create a stimulus set that has no correlations between dimensions at all using natural images. However, as long as the predictors of the model do not form a linearly dependent set, they can be disentangled in analysis by considering the unique variance explained by each. Historically, our analysis is related to studying one predictor of interest at a time, e.g., face-selective regions[36]. More recently, studies focused on decorrelating two predictors e.g., shape and animacy[31]. However, to our knowledge, none of the studies tried to decorrelate as many as the five selected dimensions of animacy. More broadly, our novel stimulus selection procedure using a genetic algorithm could be adopted to disentangle other multidimensional concepts beyond animacy.

The brain activity patterns did not fall into a small number of clusters, such as the animate and inanimate clusters observed in[1]. This does not contradict previous findings, but rather reflects the design of the stimulus set, which was optimized to reveal the ambiguities at the boundaries between the categories, far from the prototypically animate and prototypically inanimate stimuli. The Kriegeskorte et al. study is a good example of what happens when a wide range of common things are selected to define the stimulus set: many of them are low on all of the animacy dimensions studied here and many others are high on all of them. Here, by contrast, the stimulus set was designed to evenly populate a 5-dimensional space. The prototypes of animate and inanimate things fall in diametrically opposed corners of this 5-dimensional space. Those two corners are populated by very few stimuli (those that are either low on all five dimensions or high on all five dimensions). Most of the stimuli sample the unknown territory close to the boundary, which has not been explored in previous studies.

When evaluating variance explained by different dimensions it is important to keep in mind that the dimensions may not be equally represented by the stimuli. This issue can be illustrated with a conceptual experiment that aims to compare the contribution of colour and orientation in explaining early visual responses. If the presented stimuli vary in orientation by a few degrees only, whereas they vary in colour by a wide range of hues,

then we can't directly compare the contribution of orientation and colour dimensions. Since the variance of orientation would be much smaller than the variance of colour, colour will most likely contribute more to explaining the representations. A more fair comparison would be if we used all 360 degrees for orientation and all possible colour hues for colour. Unlike in the example above we use the same scale for all the dimensions of animacy (values between −10 and 10), which make the dimensions comparable to each other. However, a similar issue exists in our study in that it is impossible to equalize the distribution of values for each dimension using a natural stimulus set. We already know that some of the dimension combinations are more sampled than others. This issue does not change the interpretation of our findings but rather points at the overall issue that arises when multidimensional concepts are compared. In future studies, it will be important to validate the generalizability of the results presented here with a larger stimulus set spanning a wide range of object categories.

Despite its prominence in the neuroscience literature[3–5], the living/non-living distinction ("being alive") did not explain variance in brain representations. This finding suggests that the ventral and dorsal visual streams do not represent the results of a deeper cognitive inference process that would assess whether something is alive. Our EEG result is consistent with a magnetoencephalography (MEG) study where the "living" dimension did not explain much variance in MEG representations[23]. The fact that "looking like an animal" was the only dimension that explained significant unique variance in the EEG data may be because this dimension provides an accessible visual correlate of animacy that can be computed by the visual system. For a stimulus set where dimensions of animacy were not decorrelated, we predict that "being alive" would explain substantial variance. Our decorrelated stimulus set revealed that other dimensions underlie the responses to living things. Likely "having agency" dimension has captured some variance of the "being alive" dimension in the brain responses as the "being alive" dimension was correlated with the "having agency" dimension (Fig. 1b). Studying the interactions between the dimensions will be an important future work direction. The "being alive" dimension did explain variance in animacy ratings and similarity judgements suggesting that this dimension is present in cognition despite its lack of prevalence in the brain responses. Our result of differential representations of dimensions of animacy in brain measurement modalities and in brain and behavioural responses is consistent with previous studies that investigated a given animacy dimension of interest. For example, MEG patterns do not seem to carry information about the animate vs inanimate object category, in

contrast to fMRI[37]. Another study has shown that animate-looking (e.g., cow mug) and animate objects (e.g., cow) are dissociated in behaviour but not in the ventral visual stream[13]. Little agreement between behavioural similarity judgements and 7 T fMRI responses has also been found in[38].

Consistent with higher-cognitive contributions, the "having agency" dimension explained significant variance in higher-level visual areas, explained more variance than most other animacy dimensions later in time, and was prominent in the judgements. The observation of the "having agency" dimension explaining high-level visual representations is consistent with previous studies that showed agency representations in the fusiform gyrus (refs.[17–22,30,15–17]) and the ventral visual cortex (refs.[17–22,30]). Gobbini et al. used point-light displays as stimuli rather than images of natural objects used in this study. Beauchamp et al. and Shultz et al. used two computer-animated avatar characters. Stimuli in Throat et al. were from 40 animal categories. The number and the diversity of stimuli in our study allowed us to disentangle the dimensions of animacy experimentally with higher precision. The "having agency" dimension was least consistently judged in behavioural animacy ratings suggesting that different people have different intuitions on whether something has agency; for example, our participants were divided as to whether the robots in our stimuli had agency or not. Seeing agency (or not) in robots mirrors an ongoing debate in society and may influence how humans interact with the increasing presence of robots in their environment.

The "being unpredictable" dimension captured representations in both lower and higher-level visual cortex and earlier in time. One interpretation of this finding is that unpredictable things require attention and need to be processed early on. Humans need to know what to attend to in the visual world and if something unpredictable happens it captures attention, which enables us to stay on top of what is happening around us. This dimension of animacy has been studied only in the language domain and only in the context of natural forces[20], but we now show that "being unpredictable" also explains visual representations using a variety of stimuli. Lowder & Gordon et al. found that natural forces are processed like animate entities during online sentence processing based on an eye-tracking experiment. They propose an alternative explanation why the "being unpredictable" may be important in contrast to the attention-based explanation mentioned above. They claim that the "being unpredictable" dimension reflects a cognitive and linguistic focus on casual explanations that increase the predictability of events, which could offer an alternative theory to explain our results.

Our study did not include human-centred dimensions (human-likeness[24], humanness[23], resemblance to human faces and bodies[25], capacity for self-movement and thought rather than face presence[26]). It would be interesting to include these interpretations of animacy in a follow-up study when designing a larger stimulus set with a greater number of dimensions of animacy that we try to disentangle. Some of the dimensions tested in this study (e.g., "looking like an animal") may be correlated with the human-centred dimensions (e.g., "has a face") and exploring the relationship between the five selected dimensions of animacy and the human-centred dimensions could be a focus of future studies. Previous observations that animate versus inanimate objects can be distinguished using classifiers based on colour information[28], as well as curvature and mid-level shape features[29] inspired our control analyses that tested whether the five selected dimensions of animacy can explain unique variance over and above the low-level visual features and indeed they can. Future stimulus selection procedures using a genetic algorithm could benefit from accounting for the low-level visual features at the stimulus set selection stage.

Our study disentangled the five selected dimensions of animacy and will pave the way for future studies. For example, more work is needed to understand how exactly something quite abstract like "agency" or "unpredictability" is computed from visual stimuli. Testing alternative theories of animacy dimension computations could be addressed by comparing different classes of models in their ability to explain the data. The subsequent acquisition of higher resolution fMRI data (7 T) would provide insights into the finer-grained spatial organisation of animacy dimensions and their representations across cortical layers. Future studies may extend our approach to videos because some of the dimensions like "having mobility" may be better represented dynamically.

In summary, we disentangled the five selected dimensions of animacy using an approach for stimulus decorrelation and showed the contribution of each animacy dimension in explaining human brain representations and behavioural judgements. These dimensions captured behaviour well. A significant amount of variance in brain representations was also explained by most dimensions (with the surprising exception of "being alive"), however, to a lesser extent than in behaviour. Our results suggest that different brain regions sensitive to animacy may represent distinct dimensions, either as accessible perceptual stepping stones toward detecting whether something is alive or because they are of behavioural importance by themselves. Future studies may expand on the representation of each of the dimensions while avoiding their entanglement and may apply our dimension disentanglement approach to other multidimensional concepts.

## Methods

**Stimulus set generation: Filling animacy dimension grid combinations**. We created a grid with all possible combinations of dimensions of animacy ($2^5 = 32$). We asked participants (S = 11, mean age = 33, 6 females) to write down object category names (e.g., "humanoid robot") for each combination in the grid to obtain a list of object categories (Fig. 1a Step 1). Participants listed 100 categories, and we selected 3 images per category (total = 300 images, Fig. 1a Step 2), which formed the basis for the experiment to rate the dimensions of animacy.

**Stimulus set generation: Ratings of dimensions of animacy to generate stimulus set**. Twenty-six participants (mean age = 25, 21 females) performed animacy ratings of 300 object images through an on-line web-based interface. Participants judged each object image using a continuous scale from −10 to +10 for each dimension, e.g., −10 meant "dead" and +10 meant "alive" for the "being alive" dimension (Fig. 1a Step 3). Thirty images were repeated for a within-participant consistency measure.

**Stimulus set generation: Stimuli subset selection using a genetic algorithm**. To select a subset of 128 images for which ratings on the dimensions of animacy were maximally decorrelated we used a genetic algorithm. A genetic algorithm is an optimization method that mimics biological evolution through natural selection. Fitness was defined as minimising the maximum correlation between dimensions of animacy (Fig. 1a Step 4). We also introduced a penalty if the algorithm selected more than two stimuli from the same category (to ensure that stimuli were selected from a wide range of categories) and if the algorithm did not select at least one image of a human face and a human body (to have a reference point of object images that we know should have high ratings on the dimensions of animacy).

**Stimulus set generation: Stimuli**. All stimuli are displayed in Fig. 2a. Stimuli were 128 coloured images of real-world objects with natural backgrounds, selected from the Internet. The same set of stimuli was used in animacy ratings, similarity judgements, EEG, and fMRI experiments.

**Participants**. The same nineteen participants performed an on-line animacy ratings, similarity judgements, EEG, and fMRI experiments (mean age = 27, 13 females). Participants had normal or corrected-to-normal vision. All of them were right-handed. Before completing the experiment, participants received information about the procedure of the experiment and gave their written informed consent. All participants received monetary reimbursement or course credits for their participation. The experiments were approved by the Ethics Committee of the Department of Education and Psychology at Freie Universität, Berlin. Participants first completed the EEG and fMRI experiments, then the similarity judgements experiment, and finally the animacy ratings experiment so that they did not know

about specific dimensions of animacy tested while performing EEG, fMRI, and similarity judgements experiments.

**Animacy ratings: experimental design and task**. Participants judged each object image using a continuous scale from −10 to +10 for each animacy dimension, e.g., −10 meant "dead" and +10 meant "alive" for the "being alive" dimension (Fig. 3a). Additionally, participants performed a rating of "being animate" dimension in a similar fashion.

**Similarity judgements: experimental design and task**. We acquired pairwise object-similarity judgements for all 128 images by asking participants to perform an on-line multi-arrangement task using Meadows platform (www.meadows-research.com). During this task, object images were shown on a computer screen in a circular arena, and participants were asked to arrange the objects by their similarity, such that similar objects were placed close together and dissimilar objects were placed further apart (Fig. 5a). The multi-arrangement method uses an adaptive trial design, showing all object images on the first trial, and selecting subsets of objects with weak dissimilarity evidence for subsequent trials. To determine which stimuli to select for the next trial, the evidence weight of each stimulus had an evidence utility exponent (E = 10) applied to it, to calculate its utility if the stimulus was picked. The similarity judgements task was completed if, among all pairs of stimuli, the pair with the lowest evidence had an evidence weight higher than 0.5. The multi-arrangement method allows the efficient acquisition of a large number of pairwise similarities. We deliberately did not specify which object properties to focus on, to avoid biasing participants' spontaneous mental representation of the similarities between objects. We aimed to obtain similarity judgements that reflect the natural representation of objects without forcing participants to rely on one given dimension. However, participants were asked after having performed the task, what dimension(s) they used in judging object similarity. All participants reported arranging the images according to categorical clusters. The reports suggest that participants used a consistent strategy throughout the experiment. The method of the object similarity judgements has been described in[39], where further details can be found.

**EEG: experimental design and task**. Stimuli were presented at the centre of the screen for 500 ms, while participants performed a paper clip detection task. Stimuli were overlaid with a light grey fixation cross and displayed at a width of 4° visual angle. Participants completed 15 trials. Each image was presented twice in every trial in random order with an inter-trial interval of 1–1.1 s. Participants were asked to press a button and blink their eyes in response to a paper clip image shown randomly every 3–5 trials (mean performance 99% (+/−0.09, standard error)). These trials were excluded from the analysis.

**EEG: acquisition**. The electroencephalogram (EEG) signals were acquired using BrainVision actiCHamp EASYCAP 64 channel system at a sampling rate of 1,000 Hz. The arrangement of the electrodes followed the standard 10-20 system.

**EEG: preprocessing**. The time series were analysed with Brainstorm (http://neuroimage.usc.edu/brainstorm/). We extracted EEG patterns for each millisecond time point (from 100 ms before stimulus onset to 1000 ms after stimulus onset) for each trial. We filtered the responses between 0 and 50 Hz.

**EEG: decoding**. We performed pairwise decoding across stimuli using a support vector machine (SVM) approach[40]. For each time point, EEG signals were arranged in 64-dimensional vectors (corresponding to the 64 EEG channels), yielding M = 30 pattern vectors per time point and condition. We sub-averaged the M vectors in groups of k = 5 with random assignment, obtaining L = M/k averaged pattern vectors. This procedure was performed to reduce computational load and improve the signal-to-noise ratio. Subsequently, for each pair of conditions, we assigned L-1 averaged pattern vectors to train a linear SVM using the LibSVM implementation (www.csie.ntu.edu.tw/~cjlin/libsvm). We used the trained SVM to predict the condition labels of the left-out testing data set consisting of the Lth averaged pattern vector. This process was repeated 100 times with random assignment of the M raw pattern vectors to L averaged pattern vectors. For every time point, we assigned the average decoding accuracy to a decoding accuracy matrix.

**EEG: peak latency analysis**. We defined peaks of the decoding accuracy as time points with the maximum decoding accuracy.

**fMRI: experimental design and task**. Stimuli were presented using a rapid event-related design (stimulus duration, 500 ms) while participants performed a fixation-cross-brightness-change detection task, and their brain activity was measured with a 3 T fMRI scanner. Stimuli were overlaid with a light grey fixation cross and displayed at a width of 4° visual angle. Each image was presented once per run in random order. Each run contained 32 randomly timed null trials (null trial duration, 500 ms) without the stimulus presentation (grey square background with a fixation cross). Participants had to report a short (100 ms) change in the luminance of the fixation cross via a button press (mean performance 97% (+/−0.14,

standard error)). On average, reaction times for fixation cross trials were 0.55 s (+/−0.06 s, 2.45 s before the subsequent stimulus trial). The fixation cross was always present between stimuli or null trial presentations.

**fMRI: acquisition**. Magnetic resonance imaging was acquired using Siemens 3 T Trio with a 12-channel head coil. For structural images, we used a standard T1-weighted sequence (176 slices). The TR was 2 s, and the inter-trial-interval was 3 s. For fMRI, we conducted 9–13 runs in which 249 volumes were acquired for each participant. The number of runs varied per participant as different participants took a different number of breaks during the experiment and different amount of time was needed for them to become comfortable in the scanner. The average number of runs per participant was 10.9 (+/−0.07, standard error). The acquisition volume covered the full brain.

**fMRI: estimation of single-image activity patterns**. The fMRI data were pre-processed using SPM12 (https://www.fil.ion.ucl.ac.uk/spm/). For each participant and session separately, functional data were spatially realigned, slice-time corrected, and coregistered to the participant-individual T1 structural image. We estimated the fMRI responses to the 128 image conditions with a general linear model (GLM), which included movement parameters as nuisance terms. To obtain a t-value for each voxel and condition, GLM parameter estimates for each condition/stimulus were contrasted against a baseline. To assess the degree of the general visual stimulation we contrasted the parameter estimates for all images against the baseline.

**fMRI: definition of regions of interest**. For the ROI definition, we used a Probabilistic brain atlas[41]. Anatomical masks were reverse-normalized from MNI-space to single-participant space. For each ROI, we extracted a multivoxel pattern of activity (t-values) for each of the 128 stimuli. We included 100 most strongly activated voxels in the ROI analysis.

**fMRI: searchlight analysis**. To analyse fMRI data in a spatially unbiased approach, we performed a volume-based searchlight analysis[42] in each participant (radius of 4 voxels) with each animacy dimension RDM. We restricted the voxels included in the significance testing to visual stream areas (one-sided Wilcoxon signed-rank test).

**Construction of representational dissimilarity matrix**. We used the representational similarity analysis toolbox for animacy dimension comparison[32]. We computed response patterns (across animacy ratings, similarity judgements, EEG, and fMRI signals) for each image. We then computed response-pattern dissimilarities between images (using Euclidean distance as a metric) and placed these in a representational dissimilarity matrix (RDM). An RDM captures which distinctions among stimuli are emphasized and which are de-emphasized.

**Unique variance analysis**. We used a hierarchical general linear model (GLM) to evaluate the unique variance explained by dimensions of animacy[43]. For each animacy dimension m, the unique variance was computed by subtracting the total variance explained by the reduced GLM (excluding the dimension of interest) from the total variance explained by the full GLM. Specifically, for dimension m, we fit GLM on X = "all dimensions but m" and Y = data, then we subtract the resulting $R^2$ from the total $R^2$ (fit GLM on X = "all dimensions" and Y = data). We performed this procedure for each participant and used non-negative least squares to find optimal weights. A constant term was included in the GLM model. We performed a one-sided Wilcoxon signed-rank test to evaluate the significance of the unique variance contributed by each dimension across participants controlling the expected false discovery rate at 0.05.

**Statistics and reproducibility**. We performed inferential analyses of animacy dimension performance by correlating the animacy dimension and data RDMs using Spearman's correlation coefficient. We determined whether each of the animacy dimension RDMs was significantly related to the data RDMs using a participant-as-random-effect analysis (one-sided Wilcoxon signed-rank test). We subsequently tested for differences in animacy dimension performance between each pair of dimensions of animacy using a participant-as-random-effect analysis (two-sided Wilcoxon signed-rank test). For each analysis, we accounted for multiple comparisons by controlling the FDR at 0.05. The same nineteen participants performed an on-line animacy ratings, similarity judgements, EEG, and fMRI experiments.

**Reporting summary**. Further information on research design is available in the Nature Portfolio Reporting Summary linked to this article.

## Data availability

The datasets generated during the current study are available from the corresponding author on request. Source data for Figs. 4b, c, 5c, d, and 7a are provided in Supplementary Data 1.

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

## Acknowledgements
This research was supported by the Wellcome Trust grant [206521/Z/17/Z] awarded to KMJ; the Alexander von Humboldt Foundation postdoctoral fellowship awarded to KMJ; the German Research Council grants [CI241/1-1, CI241/3-1, CI241/7-1] awarded to RMC; and the European Research Council grant [ERC-StG-2018-803370] awarded to RMC. The images presented to the participants were sourced from https://wordpress.org/openverse/, https://www.flickr.com/ and https://images.google.com/ and are licensed under a Creative Commons license. For the purpose of open access, the author has applied a CC BY public copyright licence to any Author Accepted Manuscript version arising from this submission.

## Author contributions
K.M.J., I.C., N.K. and R.M.C. designed the experiments. K.M.J. and E.N. collected the data. J.J.F.v.d.B. helped to set up an on-line similarity judgements experiment. K.M.J. performed the analyses. K.M.J., R.M.C., and N.K. wrote the paper. All authors edited the paper. R.M.C. and N.K. supervised the work.

## Competing interests
The authors declare no competing interests.
