## [Peer Review File · Communications Biology]

Reviewers' comments:

Reviewer #1 (Remarks to the Author):

In this manuscript, the concept of "animacy" in object representations in the human brain is broken down into five related concepts and investigated using fMRI and MEG and behavioral judgements of animacy and similarity. The authors conclude that the five dimensions are represented in both the brain and behavior responses, although the specifics are complex and not always in agreement across modalities.

A strength of the manuscript is the use of a new stimulus set -- it is an interesting idea to select the stimuli with the assistance of a genetic algorithm. The methods are also clearly described, and the use of both fMRI and EEG on the same stimulus set is an additional strength.

That said, I have some serious concerns with the theoretical framing of the study and consequently with the conclusions that are drawn from the data in the context of existing work in the field. Additionally, the data at least as they are presented do not allow for drawing clear conclusions and some key control analyses needed to interpret the results and justify the conclusions drawn are currently absent. I outline these points in detail below.

Major comments:

1. Theoretical framing -- this manuscript assumes that animacy is a fundamental organizing principle in the human brain. Although this is a recognized view, there are at least two alternatives that have increasing traction and empirical evidence in support and that are not given due consideration in the current manuscript. The authors need to make it clear how their view fits in with the rest of the literature (and of course they need not agree with these two alternatives, but they need to be acknowledged). Some of these issues are touched on in a paragraph in the discussion (line 503-516), but they are missing from the introduction and framing of the study.

(i) low level statistics/visual features correlated with animacy
e.g.

- color statistics differ for animate and inanimate objects

<https://doi.org/10.1167/18.11.1>

- mid level features such as curvature or symmetry:

<https://doi.org/10.1167/17.6.20>

<https://doi.org/10.1167/17.11.10>

(ii) the presence of faces and/or similarity to humans:

e.g.

<https://doi.org/10.1523/JNEUROSCI.2628-20.2021>

<https://doi.org/10.1016/j.neuropsychologia.2022.108192>

<https://doi.org/10.1016/j.neuroimage.2020.117139>

Related to the above,

2a. "the" five dimensions of animacy are referred to (e.g. abstract, line 39, 44), but as the authors also point out, animacy is likely a multidimensional concept (line 13). There are not to my knowledge five known dimensions of animacy that are agreed upon. This phrasing needs to be changed throughout to make this important theoretical distinction clear. Similarly, the claim that "multiple dimensions commonly conflated" (line 417-418, abstract) assumes that these five dimensions are somehow fundamental, but this is not known.

2b. There is a difference between five dimensions of a concept and things that correlate with a concept. This important distinction is not made clear in the manuscript. Is it possible that the five 'dimensions' of animacy are actually just concepts that correlate with animacy? The justification for selecting the particular five dimensions seemed to be that they had been investigated previously in

the literature, but evidence for them being fundamental underlying dimensions of animacy per se is lacking.

2c. Given that the five selected dimensions do not explain much of the brain data (Fig 6, Fig 7, Supp Fig 2), is it possible that these two other explanations for animacy (faces/humanness or visual features) are also relevant? For example, "looking like an animal", the only dimension which explained unique variance in the EEG data, is likely to be highly correlated with "has a face". [also see point 5 below]

3. Stimuli -- using a genetic algorithm is an interesting approach, and I understand the motivation to "separate out" the five factors that the authors are interested in. However, the resulting stimuli are rather unusual, and several of the categories (human fetus, disembodied eyeball, person on life support) are likely to be emotional triggers or differ in other ways from the other stimuli than just their ratings on the 5 dimensions. How might this affect the interpretation of the results?

4. The fact that the stimuli were selected to be as orthogonal on the five dimensions as possible means that important interactions between dimensions may be missed, effectively the "extremes" on five highly related dimensions are being examined and this may limit generalizability. For example, only one dimension (like an animal) made a unique contribution to explaining the variance in the EEG data. This seems curious given that the stimuli were selected to be as different as possible on the five dimensions. The implications of this need to be considered further.

5. How might visual features contribute to or explain the results? This is important to rule out in order to justify claims based on the five dimensions per se. From Fig 4- several of the categories such as lightning / volcano that score high on a given dimension (e.g. unpredictability) are also very visually distinct from categories that score low on the same dimension (e.g. key, washing machine). It is notable that "being unpredictable" was reflected in both low level and high level visual cortex in the fMRI data. Similarly, the lack of explanatory power of the five dimensions in the EEG data despite good pairwise decoding performance suggests that other stimulus factors (perhaps visual features) are driving the EEG response. Is there a way to determine whether there is an effect of these dimensions per se over and above such visual features? Control analyses or consideration of these factors are absent. I also did not understand the related claim on lines 255-256.

6. Results & data visualization -- some key analyses needed to interpret the results are currently missing.

(i) despite RSA being the analysis technique of choice here, no empirical RDMS are shown for either the fMRI or EEG data. Without this, we cannot get a good feel for the brain data. Given that the dimensions explained so little of the variance in the brain data (Fig 6, Fig 7, Supp Fig 2), and the behavior and brain data did not agree (e.g. Fig 8), it seems the data are more complex than is currently presented.

(ii) Was the basic animate/animate distinction (i.e. replicating Kriegeskorte et al 2008) in VTC apparent with these stimuli? This seems an important starting place if the claim is that the stimuli reflect five dimensions of animacy. If it is not, this needs to be taken into consideration in interpreting the results.

(iii) There are no noise ceiling estimates on the RDM correlation plots for the brain data (Fig 6, Fig 7).

7. Results/Conclusion -- "We conclude that the multiple dimensions commonly conflated in the notion of animacy are distinctly reflected in different brain and behavioral responses." (abstract). However, in the EEG data only 1/5 dimensions explained unique variance and the contribution of visual features has not been addressed sufficiently in the EEG or fMRI data. Further, evidence that these five dimensions are fundamental to animacy is lacking. As it stands this claim does not seem backed up by the results as they are currently presented.

Minor:

- some in-text citations are missing from the reference list

Reviewer #2 (Remarks to the Author):

The authors examined how 5 dimensions that seemed to underlie judgements of animacy explained animacy judgements explicitly, implicitly via similarity judgments, and in brain dynamics.

The paper is interesting and quite an impressive body of work. However, I feel that because there is so much in one paper each section, especially with respect to the methods, suffers a bit. Also, there seemed to be such differences in behavior and brain (namely whether something was alive) - I was curious what the authors thought the neural mechanism underlying the behavior was?

Also, the authors need to appreciate the context of the experiment. I would have really liked to see the results replicate with another stimuli set that wasn't picked to represent dimensions of animacy. I think showing pictures of fetuses, robots, organs, puts a context to thinking about the objects presented, so I would be curious how much the results stand when having something like 120 random object categories.... or bet yet, 500. I am not convinced that the results would hold across all contexts of viewing objects.

More specific points:

1. for step 1 - in the stimulus generation, did you give the participants 100 object classes to use? Or across all 12 participants, they came up with 100 individual classes? How did you decide if participants were referring to the same object class? Or how distinct were the object classes? For example, is robot and humanoid robot two different classes?
2. Why were 29 (or 26) out of the 32 possible combinations of animacy dimensions used? What about the other three?
3. There were 128 images spanning 68 categories? Why were some categories more represented than others? Again, what is considered a category?
4. In 118 what objects were repeated? Was it the category that was repeated, the specific object, or the exact picture?
5. Were labels provided with the images? For example, for the shadow picture I am not sure if I would know to judge the shadow itself or the person creating the shadow.
6. Lines 172-175 is a bit hard to follow. It seems you are making a claim without showing data.
7. Lines 251-253 - it would be good to have data for to quantify this. All object categories?
8. line 260-262. The authors claim that when judging object similarity humans use all dimensions. But they can only make claims about the current task, with the context of the current stimuli. If with 500 different object categories that weren't focused on differentiation of agency, these dimensions may not be an organizing principle.
9. For the EEG decoding - what was the comparison - was it labeling the trial correctly out of 5 possibilities (e.g., robot vs. washing machine vs. ball vs. volcano vs. shadow)? Or was it robot vs else?
10. I would be curious taking the dimensions together in a model - when and where is the overall concept of animacy processed (irrespective of based on which dimension)? Or is that there is no one

time point/area and it is all related to the more fundamental dimensions being processed?

11. What were the correlations between dimensions?

12. For the fMRI what was the inter-trial-interval? Was there fixation the whole time between stimuli? For the null trials how long were they? Was a fixation cross presented? What was the RT for the fixation task and how close was that to the next stimulus trial?

13. Why did the number of runs vary per participant? What was the average number of runs?

14. How were the ROIs chosen? Only those ROIs that covered the 100 most strongly activated voxels? There were no strongly active voxels outside the visual stream? Was there a minimum cluster size used to be included in an ROI? I don't understand what the ROIs mean, and it would be good to have them displayed - for example, is V1 specific to V1 cortex? What about the other early visual regions? What does the 2 denote for the PHC? Did you look at posterior fusiform in comparison to more lateral aspects of LO?

15. I wonder what other things being unpredictable correlates with. What is the dimension really getting at? Does size correlate with it? Does human-ness correlate with it?

Response to the reviewers

We have made substantial revisions to address the reviewers' concerns in the current manuscript version, as outlined below. The changed text in the manuscript is in orange, addressing both reviewers' comments.

The major improvements are as follows:

- added control analyses to account for a potential confound of low-level visual features in our stimuli using an early layer of a deep convolutional neural network as a proxy of such features (AlexNet convolutional layer 1, as requested by R1)
- extensively visualized brain data as multidimensional scaling plots and colour-coding dimensions of animacy (as requested by R1)
- compared how well general animacy explains brain data in comparison to the five dimensions of animacy tested (as requested by R2)
- extended fMRI ROI analysis to 10 more brain regions and provided a visual representation of the ROI locations (as requested by R2)
- calculated and displayed noise ceilings in brain data (as requested by R2)
- added more extensive methodological details (as requested by R2)
- contextualised the study in the existing literature, more clearly explaining how our approach differs from that commonly used in the field and improved the theoretical framing of the study (as requested by R1)

Please see the response to each of the reviewers for more details.

Reviewers' comments:

Reviewer #1

Referee Expertise: Neural correlates of facial recognition

In this manuscript, the concept of "animacy" in object representations in the human brain is broken down into five related concepts and investigated using fMRI and MEG and behavioral judgements of animacy and similarity. The authors conclude that the five dimensions are represented in both the brain and behavior responses, although the specifics are complex and not always in agreement across modalities.

A strength of the manuscript is the use of a new stimulus set -- it is an interesting idea to select the stimuli with the assistance of a genetic algorithm. The methods are also clearly described, and the use of both fMRI and EEG on the same stimulus set is an additional strength.

That said, I have some serious concerns with the theoretical framing of the study and consequently with the conclusions that are drawn from the data in the context of existing work in the field. Additionally, the data at least as they are presented do not allow for drawing clear conclusions and some key control analyses needed to interpret the results and justify the conclusions drawn are currently absent. I outline these points in detail below.

We thank the reviewer for their feedback, which has helped improve the manuscript. We appreciate that the reviewer finds our new stimulus and the approach to choosing stimuli using a genetic algorithm a strong point of the manuscript and appreciates the use of both EEG and fMRI in the same stimulus set and a clear description of the methods used.

We agree that in the previous manuscript version, we did not properly contextualise the study with reference to the existing literature, and the theoretical framing was not sufficiently developed. We have made revisions to do so in the current version of the manuscript, as outlined below. We thank the reviewer for the suggested analyses that helped ground the conclusions we could draw from our study.

We have made substantial revisions to address the reviewer's concerns in the current manuscript version, as outlined below.

Major comments:

1. Theoretical framing -- this manuscript assumes that animacy is a fundamental organizing principle in the human brain. Although this is a recognized view, there are at least two alternatives that have increasing traction and empirical evidence in support and that are not given due consideration in the current manuscript. The authors need to make it clear how their view fits in with the rest of the literature (and of course they need not agree with these two alternatives, but they need to be acknowledged). Some of these issues are touched on in a paragraph in the discussion (line 503-516), but they are missing from the introduction and framing of the study.

(i) low level statistics/visual features correlated with animacy
e.g.

- color statistics differ for animate and inanimate objects

<https://doi.org/10.1167/18.11.1>

- mid level features such as curvature or symmetry:

<https://doi.org/10.1167/17.6.20>

<https://doi.org/10.1167/17.11.10>

(ii) the presence of faces and/or similarity to humans:
e.g.

<https://doi.org/10.1523/JNEUROSCI.2628-20.2021>

<https://doi.org/10.1016/j.neuropsychologia.2022.108192>

<https://doi.org/10.1016/j.neuroimage.2020.117139>

We thank the reviewer for encouraging us to explain the theoretical framework of our study in a more extensive way. In particular, we agree that there are different ways of thinking about what constitutes animacy. The revised paper discusses other dimensions related to animacy, including the presence of faces and/or similarity to humans and cites the studies that the reviewer suggested (Ritchie et al., 2021, Proklova & Goodale, 2020 and Contini et al. 2019).

However, our study goes beyond the standard approach in the literature, where a given study typically only investigates one dimension of interest and tries to show that this

dimension is represented. Different dimensions are studied in isolation, making it difficult to assess whether results reflect confounding dimensions and what different dimensions' relative contributions are. Our study filled this gap by studying several animacy dimensions that are commonly confounded using a stimulus set that decorrelates them. It would be ideal to include all the dimensions that have ever been considered in the literature, but this is difficult given the limited number of stimuli that we can present in a single study. Some of these dimensions have also been proposed after the study had been designed and data collected. We chose five dimensions based on the literature, but do not claim that these are the only ones.

The revised manuscript clearly acknowledges the limitations of our study. We have also added information about the importance of the low-level and mid-level visual features in animacy representations in the introduction, results, and discussion sections. We added additional Supplementary Figures (2, 7, and 16) to address the contribution of low-level visual features in explaining brain representations in contrast to the five selected dimensions as discussed in detail in point 5. We have pasted the added text below:

The relevant paragraph from the introduction section:

“Apart from the abovementioned dimensions of animacy, several other human-centred interpretations of animacy have been proposed. Recently reported animacy-related concepts that explain variance in the ventral visual stream fMRI measurements are human-likeness (Rosenthal-von der Pütten et al., 2019), humanness (Contini et al., 2019), resemblance to human faces and bodies (Ritchie et al., 2021), and capacity for self-movement and thought rather than face presence (Proklova & Goodale, 2020). Another similar concept is the animacy continuum, where objects are perceived as more animate when they are more similar to humans (e.g., images of monkeys would be perceived as more animate than insects, even though both species belong to the animal category; Connolly et al., 2012). In addition to the high-level dimensions, low-level visual features correlate with animacy (Long et al., 2017; Rosenthal et al., 2018; Schmidt et al., 2017). Colour statistics (Rosenthal et al., 2018), curvature (Long et al., 2017), and mid-level shape features (Schmidt et al., 2017) can be used to classify whether an object is animate or inanimate.”

The relevant paragraph from the discussion section:

“Our study did not include human-centred dimensions (human-likeness (Rosenthal-von der Pütten et al., 2019), humanness (Contini et al., 2019), resemblance to human faces and bodies (Ritchie et al., 2021), capacity for self-movement and thought rather than face presence (Proklova & Goodale, 2020, animacy continuum)). It would be interesting to include these interpretations of animacy in a follow-up study when designing a larger stimulus set with a greater number of dimensions of animacy that we try to disentangle. Previous observations that animate versus inanimate objects can be distinguished using classifiers based on colour information (Rosenthal et al., 2018), as well as curvature (Long et al., 2017) and mid-level shape features (Schmidt et al., 2017) inspired our control analyses that tested whether the five selected dimensions of animacy can explain unique variance over and above the low-level visual features and indeed they can. Future stimulus selection procedures using a genetic algorithm could benefit from accounting for the low-level visual features at the stimulus set selection stage.”

Related to the above,

2a. "the" five dimensions of animacy are referred to (e.g. abstract, line 39, 44), but as the authors also point out, animacy is likely a multidimensional concept (line 13). There are not to my knowledge five known dimensions of animacy that are agreed upon. This phrasing needs to be changed throughout to make this important theoretical distinction clear. Similarly, the claim that "multiple dimensions commonly conflated" (line 417-418, abstract) assumes that these five dimensions are somehow fundamental, but this is not known.

We agree with the reviewer that other dimensions have been implicated in animacy. The phrase "the five dimensions" was intended to refer to the five selected for investigation in our study. To avoid any misreading, following the reviewer's suggestion, we have changed the language in the manuscript to ensure that the five dimensions tested are not explicitly or implicitly suggested to be the only dimensions of animacy that might be relevant. We specifically replaced "the five dimensions" with "five dimensions related to animacy" and "the five selected dimensions" throughout the manuscript. We agree that the claim that "multiple dimensions commonly conflated" may seem confusing; we have changed the conclusion of the abstract. Below are the modified sentences from the abstract, but we changed the wording throughout the manuscript.

"Here, we investigate the importance of five dimensions related to animacy ("being alive", "looking like an animal", "having agency", "having mobility", and "being unpredictable") in brain (fMRI, EEG) and behaviour (property and similarity judgements) of 19 participants. We use a stimulus set of 128 images, optimized by a genetic algorithm to disentangle the five chosen dimensions."

"Our results suggest that different brain regions sensitive to animacy may actually represent distinct dimensions, either as accessible perceptual stepping stones toward detecting whether something is alive or because they are of behavioural importance in their own right."

2b. There is a difference between five dimensions of a concept and things that correlate with a concept. This important distinction is not made clear in the manuscript. Is it possible that the five 'dimensions' of animacy are actually just concepts that correlate with animacy? The justification for selecting the particular five dimensions seemed to be that they had been investigated previously in the literature, but evidence for them being fundamental underlying dimensions of animacy per se is lacking.

We thank the reviewer for pointing out the subtle difference between dimensions of concept and dimensions that correlate with a concept. We now address this point explicitly in the introduction:

"Dimensions that define a concept depend on the chosen definition of the concept. Dictionary definitions of animate rely heavily on the dimension of being alive (animate: "living; having life", Oxford Advanced Learner's Dictionary). However, being alive is a difficult-to-assess latent property of a thing, so it seems possible that a perceptual system might represent more accessible dimensions that are correlated with being alive, even if being alive were ultimately

the behaviourally important property. In addition, the more accessible related properties (such as those mentioned above) may be of behavioural importance in their own right. We are not concerned here with the philosophical and semantic questions of animacy, but with the empirical question of which of several related and commonly conflated dimensions are represented in particular brain regions and in behavioural judgements.”

2c. Given that the five selected dimensions do not explain much of the brain data (Fig 6, Fig 7, Supp Fig 2), is it possible that these two other explanations for animacy (faces/humanness or visual features) are also relevant? For example, "looking like an animal", the only dimension which explained unique variance in the EEG data, is likely to be highly correlated with "has a face". [also see point 5 below]

We agree that other dimensions of animacy may explain brain representations. We cannot test this directly in this study as our stimulus set was not decorrelated on the other dimensions mentioned by the reviewer, and we don't have ratings on these dimensions. As a response to the reviewer's comment, we have now discussed that the dimensions of animacy tested in our study may be correlated with some of the other dimensions like the example the reviewer has brought up, i.e. the "looking like an animal" dimension may be correlated with the "has a face" dimension. We performed additional analyses to test the contribution of the low-level visual features and discussed these results in point 5.

“Some of the dimensions tested in this study (e.g., "looking like an animal") may be correlated with the human-centred dimensions (e.g., has a face) and exploring the relationship between the five selected dimensions of animacy and the human-centred dimensions could be a focus of future studies.”

3. Stimuli -- using a genetic algorithm is an interesting approach, and I understand the motivation to "separate out" the five factors that the authors are interested in. However, the resulting stimuli are rather unusual, and several of the categories (human fetus, disembodied eyeball, person on life support) are likely to be emotional triggers or differ in other ways from the other stimuli than just their ratings on the 5 dimensions. How might this affect the interpretation of the results?

We are happy to hear that the reviewer finds our approach of using the genetic algorithm interesting and that the reviewer understands why we decided to go with this approach. Because we wanted to obtain all possible combinations of the five dimensions of interest, we had no choice but to include some unusual stimuli. (Indeed, using “usual” stimuli is exactly what led to the confounding of distinct dimensions in previous studies.) However, none of the participants mentioned that they found any stimuli upsetting after performing the experiment. The stimuli that the reviewer refers to as unusual constitute a small percentage of the stimulus set (7.8 %, 10 out of 128 stimuli, considering that stimuli such as human fetus, disembodied eyeball, person on life support, heart can be considered emotional triggers). As the unusual stimuli constitute only a small percentage of our stimulus set, we do not think this would affect our results' interpretation. However, we have now discussed this point raised by the reviewer in the manuscript providing the information written above and discussing that future studies may consider factoring in stimuli affect ratings when designing the stimulus set.

“A small percentage of stimuli can be considered as unusual (7.8 %, 10 out of 128 stimuli, considering that stimuli such as human fetus, disembodied eyeball, person on life support, heart can be considered as emotional triggers). As the unusual stimuli constitute only a small percentage of our stimulus set we do not think that they would affect the interpretation of our results. None of the participants mentioned that they found any of the stimuli upsetting after performing the experiment. However, future studies may benefit from including affect ratings alongside the dimensions of animacy ratings.”

4. The fact that the stimuli were selected to be as orthogonal on the five dimensions as possible means that important interactions between dimensions may be missed, effectively the "extremes" on five highly related dimensions are being examined and this may limit generalizability. For example, only one dimension (like an animal) made a unique contribution to explaining the variance in the EEG data. This seems curious given that the stimuli were selected to be as different as possible on the five dimensions. The implications of this need to be considered further.

It is not clear to us what the reviewer means by “interactions between dimensions may be missed” when stimuli are selected to decorrelate dimensions. It seems to us that interactions are missed (cannot be investigated at all) when a natural stimulus distribution is chosen that confounds distinct dimensions.

We agree that it is curious that despite the strong variation along each of the five dimensions only “looking like an animal” explained significant unique variance in the EEG data. We now interpret this key result in the discussion and the abstract:

“The fact that “looking like an animal” was the only dimension that explained significant unique variance in the EEG data may be because this dimension provides an accessible visual correlate of animacy that can be computed by the visual system.”

“The “looking like an animal” dimension was the only dimension that explained significant unique variance in the EEG data. One interpretation is that “looking like an animal” provides an accessible visual correlate of animacy that can be computed by the visual system.”

We have elaborated on generalizability more in the revised version of the discussion:

“In future studies, it will be important to validate the generalizability of the results presented here with a larger stimulus set spanning a wide range of object categories.”

5. How might visual features contribute to or explain the results? This is important to rule out in order to justify claims based on the five dimensions per se. From Fig 4- several of the categories such as lightning / volcano that score high on a given dimension (e.g. unpredictability) are also very visually distinct from categories that score low on the same dimension (e.g. key, washing machine). It is notable that "being unpredictable" was reflected in both low level and high level visual cortex in the fMRI data. Similarly, the lack of explanatory power of the five dimensions in the EEG data despite good pairwise decoding performance suggests that other stimulus factors (perhaps visual features) are driving the EEG response. Is there a way to determine whether there is an effect of these dimensions per se over and above such

visual features? Control analyses or consideration of these factors are absent. I also did not understand the related claim on lines 255-256.

We agree that considering how low-level visual features can explain our results is an important analysis that we have now included. We note that four of the five dimensions (all except “being alive”) do explain variance in the EEG (although only “looking like an animal” explained significant unique variance).

To address the reviewer’s important concern about the effect of the five animacy dimensions tested over and above the visual features, we performed a unique variance analysis for both EEG data and the fMRI ROI analysis as now described in the revised version of the results:

“We wanted to check whether the unique variance explained by the “looking like an animal” dimension in the EEG data can be related to low-level visual features or whether this dimension explains unique variance over and above the variance explained by low-level features. We therefore included the first convolutional layer of AlexNet as a model of low-level visual features in the unique variance analysis in addition to the five dimensions of animacy studied here. We observed that our result of “looking like an animal” explaining a significant amount of the unique variance still holds. “Looking like an animal” explains the unique variance not explained by either the other four dimensions or the first convolutional layer of AlexNet (Supplementary Figure 7).”

“We also wanted to test whether the five dimensions of animacy studied here explained variance over and above low-level visual features in the fMRI ROI analysis. As for the EEG analysis, we therefore included the first convolutional layer of AlexNet along with the five dimensions of animacy in the unique variance analysis. Similarly to behavioural and EEG results, selected dimensions of animacy explained variance over and above low-level visual features of AlexNet’s first convolutional layer (Supplementary Figure 16). As expected, the unique contribution of the first convolutional layer of AlexNet was greater in the early and intermediate visual cortex in comparison to the high-level visual cortex (Supplementary Figure 16).”

The claim on lines 255-256 stated, “If we assume that the similarity judgements are based only on the similarity between low-level visual features, the dimensions of animacy should not explain a large fraction of the variance.” We agree that some more unpacking is needed of what we mean. We have now expanded on it in the revised version of the results:

“If the low-level features were the only ones that participants used in similarity judgements object arrangements, then we would see on the MDS that objects are arranged by, for example, colour or shape, but this is not what we observe (Figure 5b).”

To directly test that, we performed the same analysis as we did for the EEG and fMRI ROI data explained above (unique variance analysis in the context of the dimensions of animacy and the first convolutional layer of AlexNet) in the revised version of the results:

“It is important to consider the effect of low-level visual features on the interpretation of these results and test whether the five animacy dimensions studied here explain unique variance over and above low-level visual features in similarity judgements. We included the first

convolutional layer of the deep neural network AlexNet (Krizhevsky et al., 2012) as a model of low-level visual features in this analysis. We observed that each of the animacy dimensions explained a significant amount of unique variance over and above the variance explained by low-level visual features and the other dimensions (Supplementary Figure 2).”

6. Results & data visualization -- some key analyses needed to interpret the results are currently missing.

We agree with the reviewer’s point that in addition to the inferential analyses, descriptive data visualisation is important for the purpose of discovering unexpected structure. We have now added multidimensional scaling (MDS) plots based on RDMs (including colour-coded MDS based on the five dimensions of animacy) and inferential analyses plots with noise ceiling estimates for the fMRI and EEG data.

(i) despite RSA being the analysis technique of choice here, no empirical RDMs are shown for either the fMRI or EEG data. Without this, we cannot get a good feel for the brain data. Given that the dimensions explained so little of the variance in the brain data (Fig 6, Fig 7, Supp Fig 2), and the behavior and brain data did not agree (e.g. Fig 8), it seems the data are more complex than is currently presented.

We agree that inspecting the RDMs gives valuable insights into the structure of the representations present in the data that complements the inference based on the RSA model comparison, which is what we focused on in this study. However, inspecting the RDMs is helpful only when there is a structure that lends itself to being laid out along one dimension to define the ordering of the stimuli that needs to be chosen to plot an RDM. The simple nested categorical structure of Kriegeskorte et al. 2008 is a case in point, where an ordering can capture the major categorical divisions. In the present study, by contrast, the stimulus set is designed to evenly populate a 5-dimensional space. In this context inspecting the RDMs is not helpful, unless each region’s RDM is plotted five times, using each of the 5 dimensions to define the order. We have added multidimensional scaling (MDS) visualizations based on RDMs. The revised paper uses MDS plots for the similarity judgements data in figure 5b, for selected time points of the EEG representations (Supplementary Figures 5 and 6), and for the fMRI ROIs (Supplementary Figures 13 and 14). We have added the description of this analysis to the results section and pasted the relevant text below. We have also discussed the usefulness of displaying such data for interpreting the results following the reviewer’s comment.

“To gain intuition of how well the five dimensions of animacy studied here separated representations in the similarity judgements, we have displayed MDS plots colour-coded according to binary animacy dimensions (e.g., “being alive” with one colour of dots and “not being alive” with another colour of dots, Supplementary Figure 1). All dimensions separated the stimuli well, with each dimension revealing different divisions between stimuli (Supplementary Figure 1).”

“To explore the structure of the representations, we displayed MDS plots for the selected timepoints: 0 ms - when the stimulus was just displayed, 100 ms - when the decoding accuracy started to go up, 200 ms - peak decoding accuracy, and 300 ms - when the decoding accuracy started to drop. As expected, no structure was visible at the stimulus onset (0 ms,

Supplementary Figure 5a). At 100 ms, human and humanoid and animal robot faces were grouped together, as well as forces of nature (Supplementary Figure 5b). At 200 ms, faces were still grouped together, however, faces of a human and robots were represented further away from each other (Supplementary Figure 5c). At 300 ms, we observed similar clusters to those present at 100 ms (Supplementary Figure 5d). We have also displayed MDS plots colour-coded according to binary animacy dimensions. These purely descriptive results suggest that there is a separation of the stimuli that more strongly reflects some dimensions of animacy than others (e.g., “being unpredictable” had some level of separation in contrast to “being alive” which did not). The descriptive MDS results are consistent with the amount of variance these dimensions explained in the inferential RSA analysis described below.”

“To gain intuition about the structure of the representations, we displayed MDS plots for the ROIs. As expected, the stimuli in early visual regions were grouped by shapes and colours, whereas we could see clusters of faces and forces of nature in higher-level visual regions (Supplementary Figure 13). In addition, the colour-coded MDS plot for PHC2 based on the binary dimensions of animacy revealed mild categorical structure for some dimensions (e.g., “looking like an animal”, Supplementary Figure 14).”

(ii) Was the basic animate/inanimate distinction (i.e. replicating Kriegeskorte et al 2008) in VTC apparent with these stimuli? This seems an important starting place if the claim is that the stimuli reflect five dimensions of animacy. If it is not, this needs to be taken into consideration in interpreting the results.

We also do not see strong clustering of stimuli into a small number of categories as observed in Kriegeskorte et al. (2008). We thank the reviewer for bringing up this important point, which we now address in the discussion:

“The brain activity patterns did not fall into a small number of clusters, such as the animate and inanimate clusters observed in Kriegeskorte et al. (2008). This does not contradict previous findings, but rather reflects the design of the stimulus set, which was optimized to reveal the ambiguities at the boundaries between the categories, far from the prototypically animate and prototypically inanimate stimuli. The Kriegeskorte et al. (2008) study is a good example of what happens when a wide range of common things are selected to define the stimulus set: many of them are low on all of the animacy dimensions studied here and many others are high on all of them. Here, by contrast, the stimulus set was designed to evenly populate a 5-dimensional space. The prototypes of animate and inanimate things fall in diametrically opposed corners of this 5-dimensional space. Those two corners are populated by very few stimuli (those that are either low on all five dimensions or high on all five dimensions). Most of the stimuli sample the unknown territory close to the boundary, which has not been explored in previous studies.”

(iii) There are no noise ceiling estimates on the RDM correlation plots for the brain data (Fig 6, Fig 7).

Following the reviewer’s suggestion, we have displayed the noise ceiling for the RDM correlation plots for EEG (Supplementary Figure 1) and fMRI ROI (Supplementary Figure 7) analyses and we referred to these additional plots in the revised manuscript.

“The selected dimensions of animacy do not reach the noise ceiling, indicating that it leaves unexplained some of the variance that is reliable across individual observers.”

“We found that most animacy dimensions explained a significant amount of variance in EEG recordings (Figure 6b, Supplementary Figure 3); however, some dimensions explained variance at slightly different times.”

“We first evaluated the contribution of each animacy dimension in ROIs across the ventral: visual area 1 (V1v), ventral occipital cortex 2 (VO2), parahippocampal cortex 2 (PHC2) and dorsal: visual area 1 (V1d), lateral occipital cortex 2 (LO2), TO2 visual streams (Figure 7a, Supplementary Figure 9).”

7. Results/Conclusion -- "We conclude that the multiple dimensions commonly conflated in the notion of animacy are distinctly reflected in different brain and behavioral responses." (abstract). However, in the EEG data only 1/5 dimensions explained unique variance and the contribution of visual features has not been addressed sufficiently in the EEG or fMRI data.

This sentence was indeed not clear. We meant, not that each dimension is equally, but separately reflected in brain and behavioural responses, but that they are reflected to different degrees. We have clarified the abstract to state the concrete results and edited the overall conclusion sentence to clarify this.

“Our results suggest that different brain regions sensitive to animacy may actually represent distinct dimensions, either as accessible perceptual stepping stones toward detecting whether something is alive or because they are of behavioural importance in their own right.”

Further, evidence that these five dimensions are fundamental to animacy is lacking. As it stands this claim does not seem backed up by the results as they are currently presented.

We agree that it is not warranted to claim that the five dimensions we chose to study are “fundamental to animacy”. It is indeed unclear what that would mean and whether empirical evidence could back such a claim. In the revision, we have carefully made sure that we do not either use such language or indirectly suggest that the five dimensions we study are the only relevant ones or jointly fundamental to the concept of animacy.

Minor:

- some in-text citations are missing from the reference list

We thank the reviewer for pointing out the inconsistency of the reference list. We had a technical issue with the Zotero plugin (references software). We have now made sure that all references are present in the reference list.

Reviewer #2

Referee expertise: Neural correlates of visual perception

The authors examined how 5 dimensions that seemed to underlie judgements of animacy explained animacy judgements explicitly, implicitly via similarity judgments, and in brain dynamics.

The paper is interesting and quite an impressive body of work. However, I feel that because there is so much in one paper each section, especially with respect to the methods, suffers a bit.

We appreciate that the reviewer finds the manuscript interesting and is impressed by the work presented in the manuscript.

It's indeed not easy to describe all the details of the methodology when two imaging modalities and behavioural methods are combined in the same paper, and we did not provide sufficient details on the methods in the previous manuscript version. We have added all the methodological details that the reviewer asked for in the revised version of the manuscript (stimulus set generation, correlations between dimensions, EEG decoding, fMRI acquisition, ROI definition).

Also, there seemed to be such differences in behavior and brain (namely whether something was alive) - I was curious what the authors thought the neural mechanism underlying the behavior was?

As the reviewer pointed out, there were differences between the behavioural and neural representations, and now we have added more analyses to try to understand the nature of these differences (including accounting for low-level features and visually inspecting the organisation of stimuli and the dimensions of animacy divisions in both behavioural and neural data using multidimensional scaling plots).

Also, the authors need to appreciate the context of the experiment. I would have really liked to see the results replicate with another stimuli set that wasn't picked to represent dimensions of animacy. I think showing pictures of fetuses, robots, organs, puts a context to thinking about the objects presented, so I would be curious how much the results stand when having something like 120 random object categories.... or bet yet, 500. I am not convinced that the results would hold across all contexts of viewing objects.

We share the reviewer's interest in natural, randomly sampled stimulus sets. These provide the backbone of most of our studies (e.g. Kriegeskorte et al. 2008). However, this paper addresses a set of questions that does not lend itself to this approach. The questions about the particular five dimensions of animacy that we set out to address here cannot be powerfully investigated without a stimulus set that is either huge or designed to disentangle the dimensions. It is precisely the unusual stimuli used here that enable us to explore the unknown territory between the animate category and the inanimate category. We do hope that future efforts leveraging many thousands of natural images will replicate some of our findings here.

We have made substantial revisions to address the reviewer's concerns in the current manuscript version, as outlined below.

More specific points:

1. for step 1 - in the stimulus generation, did you give the participants 100 object classes to use? Or across all 12 participants, they came up with 100 individual classes? How did you decide if participants were referring to the same object class? Or how distinct were the object classes? For example, is robot and humanoid robot two different classes?

We thank the reviewer for highlighting that more details about the object classes are needed. There was no ambiguity in terms of different names of a given object class, so we did not need to decide whether participants were referring to the same object class or not. We now included the requested details in the revised results and methods:

“First, we created an animacy dimension grid where we asked participants to fill freely in the names of the objects fulfilling each animacy dimension combination to then find images that satisfy combinations of dimensions of animacy (29 out of 32 possible combinations).”

“Participants came up with 100 classes in total, and the participants were not given any object classes to use by the experimenters (Supplementary Table 1).”

“The object categories were distinct (e.g., there were different types of robots and, therefore, two different object categories, “humanoid robot” and “animal robot”, were included).”

2. Why were 29 (or 26) out of the 32 possible combinations of animacy dimensions used? What about the other three?

We thank the reviewer for bringing to our attention that we did not describe why not all the 32 possible combinations of animacy dimensions were used. We added the explanation in the results section of the revised manuscript:

“The object names provided by the subjects did not cover all 32 combinations, which is why 29 combinations out of 32 were included in step 1 of the stimulus selection procedure.”

“Among 29 dimension combinations for which participants provided object names, three were not selected by the GA. This is because the objective of the GA was to minimise the maximum correlations between dimensions, and some dimension combinations were not optimal to be chosen.”

3. There were 128 images spanning 68 categories? Why were some categories more represented than others? Again, what is considered a category?

A category was an object label falling in one of the 32 orthants of the 5-dimensional space of animacy provided by subjects in step 1 of the stimuli selection procedure. We provided the requested details in the results:

“The GA did not choose some of the images representing object names from the initial 100 object names listed in step 1, as the dimensions of animacy ratings on these images were not optimal for the GA objective.”

“The GA was allowed to choose a maximum of two different images representing different objects from a given category (e.g., two different animal robots) if this selection contributed to an optimal GA solution.”

4. In 118 what objects were repeated? Was it the category that was repeated, the specific object, or the exact picture?

Object categories that were repeated included, e.g., animal robot, vulcano, and person on life support, as evidenced in Figure 2a. As specified in point 3 above, two pictures from the same category were allowed to be selected by the genetic algorithm. It was a category that was allowed to be repeated, not a specific object and not a specific picture. We have already included this information in the methods section of the previous manuscript version and pasted it below for the reviewer’s convenience:

“We also introduced a penalty if the algorithm selected more than two stimuli from the same category (to ensure that stimuli were selected from a wide range of categories) and if the algorithm did not select at least one image of a human face and a human body (to have a reference point of object images that we know should have high ratings on the dimensions of animacy).”

5. Were labels provided with the images? For example, for the shadow picture I am not sure if I would know to judge the shadow itself or the person creating the shadow.

To answer the reviewer’s question, the labels were not provided with the images. We wanted this to be a visual task, not biased by semantic labels that may influence subjects’ behavioural judgements and brain representations. We thank the reviewer for pointing out the confusion related to this particular example of a picture of a human shadow, which we discussed now in the results section:

“While participants were asked to judge what is represented on images (“human shadow”), it might not be apparent whether to judge the shadow itself or the person creating the shadow.”

6. Lines 172-175 is a bit hard to follow. It seems you are making a claim without showing data.

The reviewer refers to the sentence “In contrast, images with high ratings did differ depending on a dimension tested (e.g., stimuli judged as the most unpredictable being humans and forces of nature, in contrast to humans and robots judged as having the most agency), proving that indeed these dimensions capture different aspects of animacy perception.” The reviewer rightly pointed out that we did not include the reference to the figure that supports this observation. We have now included the reference to Figure 4A at the end of this sentence. Figure 4a supports our claim that the reviewer refers to where on

the right panel of the figure, the reader can see the most highly rated images on each of the animacy dimensions. So indeed, dimensions differ in terms of what they represent.

“In contrast, images with high ratings did differ depending on a dimension tested (e.g., stimuli judged as the most unpredictable being humans and forces of nature, in contrast to humans judged as having the most agency), proving that indeed these dimensions capture different aspects of animacy perception (Figure 4a, right panel).”

7. Lines 251-253 - it would be good to have data for to quantify this. All object categories?

The reviewer refers to the sentence, “Some unpredictable objects were also grouped together: geysers and game machines, or volcanos and waves. As a sanity check, images depicting the same object were grouped together, such as two pictures of flowers or wheels.”

We can see the data supporting this claim on the MDS plot in Figure 5b, as objects close together in the MDS layout are more similarly represented. We have added the reference to this figure at the end of the abovementioned sentence to clarify what plot this statement refers to. We also performed an additional analysis where we colour-coded each of the animacy dimensions in the MDS plots (Supplementary Figure 1).

“Some unpredictable objects were also grouped together: geysers and game machines, or volcanos and waves (Figure 4a). As a sanity check, images that depict the same object were grouped together, for example, two pictures of flowers or wheels (Figure 4a). To gain intuition of how well the five dimensions of animacy studied here separated representations in the similarity judgements, we have displayed MDS plots colour-coded according to binary animacy dimensions (e.g., “being alive” with one colour of dots and “not being alive” with another colour of dots, Supplementary Figure 1). All dimensions separated the stimuli well, with each dimension revealing different divisions between stimuli (Supplementary Figure 1).”

8. line 260-262. The authors claim that when judging object similarity humans use all dimensions. But they can only make claims about the current task, with the context of the current stimuli. If with 500 different object categories that weren't focused on differentiation of agency, these dimensions may not be an organizing principle.

The reviewer refers to the sentence, “This result means that when judging object similarity, humans use all dimensions of animacy with the emphasis on agency.”

As the reviewer pointed out, the conclusions of this study, as in all experimental studies, need to be considered in the context of the stimulus set used. We think that if a larger number of stimuli was studied, then the five dimensions of animacy would still explain variance in behaviour and brain representations, but this would need to be empirically tested. The reviewer mentions an important point that the generalisation of the conclusions should be verified in a larger stimulus set. However, this interesting experiment goes beyond the scope of this study. As the reviewer pointed out in one of their initial comments, the current study is already extensive with the current set of stimuli and behavioural and brain imaging techniques used. We discuss the reviewer’s suggestion as an interesting direction for future studies in the discussion section of the manuscript. As a side note, our stimulus set

focused on decorrelation of all animacy dimensions, not only agency as the reviewer eluded in their comment.

“In future studies, it will be important to validate the generalizability of the results presented here with a larger stimulus set spanning a wide range of object categories.”

9. For the EEG decoding - what was the comparison - was it labeling the trial correctly out of 5 possibilities (e.g., robot vs. washing machine vs. ball vs. volcano vs. shadow)? Or was it robot vs else?

We should have stated the information about decoding more explicitly. We performed pairwise image decoding and plotted the average across all decoding pair combinations. Following the reviewer’s question, we have added details about the stimuli decoding in the figure caption (Figure 6a) and manuscript text. During the EEG experiment, subjects did not perform any labelling task (neither the dimensions of animacy or object categories), but they performed an orthogonal paperclip detection task, so they were not biased towards either the selected dimensions of animacy or object categories.

“We performed pairwise decoding across stimuli using a support vector machine (SVM) approach (Cichy et al., 2014).”

“We performed pairwise stimuli decoding using a support vector machine approach and we could decode images in the stimulus set to a high decoding accuracy (62%) in a long time window (between 43 and 1000 ms after stimulus onset, Figure 6a).”

“Figure 6. Dimensions of animacy and EEG time course.

a. Mean decoding curve across participants (pairwise stimuli decoding using a support vector machine approach).”

10. I would be curious taking the dimensions together in a model - when and where is the overall concept of animacy processed (irrespective of based on which dimension)? Or is that there is no one time point/area and it is all related to the more fundamental dimensions being processed?

We thank the reviewer for this analysis suggestion. We have collected the general animacy rating data. In addition to the five dimensions of animacy the subjects rated, they also rated general animacy (“being animate”). Following the reviewer’s suggestions, we correlated these overall animacy ratings with the fMRI ROIs and EEG data. We discussed these findings in the results section of the revised manuscript.

“To investigate how the five dimensions of animacy studied here relate to the general animacy, we correlated the general animacy ratings alongside the dimensions of animacy with the EEG RDMs. We observed that the “being animate” ratings explained a significant amount of variance in the EEG responses, similar in magnitude and timing to the variance explained by the dimensions of animacy tested (Supplementary Figure 4).”

“After examining the contribution of general “being animate” ratings to the fMRI ROI representations, we did not see that “being animate” explained a significant amount of variance in fMRI data (Supplementary Figure 10).”

11. What were the correlations between dimensions?

The correlations between the dimensions have already been presented in Supplementary Figure 1 of the manuscript and discussed in the text. However, as eluded by the reviewer, it is a piece of important information, and we have added the correlation matrices before and after the selection of the stimuli by the genetic algorithm to the main manuscript figures (Figure 1b).

“Pairwise correlation between animacy dimensions for the randomly selected 128 stimuli (left) and the 128 stimuli selected by the genetic algorithm (right) in behavioural ratings.” (was already present in the previous version of the manuscript)

“The maximum correlation between dimensions in the stimulus set was 0.36. This result was better than when randomly selecting the stimuli 10,000 times without optimization (maximum correlation between dimensions = 0.64), proving that our novel stimulus selection procedure was successful. The pairwise correlations between animacy dimensions for the 128 stimuli selected by GA and for the 128 stimuli selected randomly are represented in Figure 1b.” (was already present in the previous version of the manuscript, but now we have moved correlation matrices from the Supplementary Figures to the main manuscript Figures)

12. For the fMRI what was the inter-trial-interval? Was there fixation the whole time between stimuli? For the null trials how long were they? Was a fixation cross presented? What was the RT for the fixation task and how close was that to the next stimulus trial?

We agree with the reviewer that more details on the fMRI data acquisition are needed, and we provided more details outlined below in the methods section of the revised manuscript:

“Each run contained 32 randomly timed null trials (null trial duration, 500 ms) without the stimulus presentation (grey square background with a fixation cross). Participants had to report a short (100 ms) change in the luminance of the fixation cross via a button press (mean performance 97% (+/- 0.14, standard error)). On average, reaction times for fixation cross trials were 0.55 s (+/- 0.06 s, 2.45 s before the subsequent stimulus trial). The fixation cross was always present between stimuli or null trial presentations.”

“The TR was 2 s, and the inter-trial-interval was 3 s.”

13. Why did the number of runs vary per participant? What was the average number of runs?

The number of runs varied per participant as different participants took a different number of breaks during the experiment and took different amounts of time to become comfortable in the scanner. We have added the requested information to the methods section:

“The number of runs varied per participant as different participants took a different number of breaks during the experiment and different amount of time was needed for them to become comfortable in the scanner. The average number of runs per participant was 10.9 (+/- 0.07, standard error).”

14. How were the ROIs chosen? Only those ROIs that covered the 100 most strongly activated voxels? There were no strongly active voxels outside the visual stream? Was there a minimum cluster size used to be included in an ROI? I don't understand what the ROIs mean, and it would be good to have them displayed - for example, is V1 specific to V1 cortex? What about the other early visual regions? What does the 2 denote for the PHC? Did you look at posterior fusiform in comparison to more lateral aspects of LO?

We agree with the reviewer that more information about the ROI location is needed. The ROIs were chosen based on a Probabilistic brain atlas (Wang et al., 2015), as described in the methods section. Within each ROI, we selected the 100 most strongly activated voxels, as described in the methods section. Searchlight analysis revealed that no strongly active voxels outside the visual stream passed the significance threshold. We did not use a minimum cluster size. Following the reviewer's suggestion, we have displayed the ROI location in Supplementary Figure 6. V1v is ventral V1, and V1d is dorsal V1. We have specified ROI names in the previous version of the manuscript and pasted the relevant text below for the reviewer's convenience. Following the reviewer's suggestion, we have examined all visual regions identified in Wang's 2015 brain atlas at different levels of the visual hierarchy and added these analyses as Supplementary Figures 11 and 12 and discussed in the result section of the revised manuscript:

“We first evaluated the contribution of each animacy dimension in ROIs across the ventral: visual area 1 (V1v), ventral occipital cortex 2 (VO2), parahippocampal cortex 2 (PHC2) and dorsal: visual area 1 (V1d), lateral occipital cortex 2 (LO2), TO2 visual streams (Figure 7a, Supplementary Figure 7). To define ROIs, we used a Probabilistic brain atlas (Wang et al., 2015). The locations of the examined ROIs are presented in Supplementary Figure 8.”

“For completeness, we performed the same analysis for all regions in ventral (V1v, V2v, hV4, VO1, VO2, PHC1, PHC2) and dorsal (V1d, V2d, V3d, V3a, V3b, LO1, LO2, TO1, TO2) visual streams. We observed that the dimensions of animacy studied here explain variance in different brain regions to different extents, and more variance is explained by the dimensions of animacy in higher-level visual cortex (Supplementary Figures 11 and 12).”

15. I wonder what other things being unpredictable correlates with. What is the dimension really getting at? Does size correlate with it? Does human-ness correlate with it?

We, unfortunately, cannot test whether the “being unpredictable” dimension correlates with object size or humanness as we do not have these other dimensions' ratings. We did not create a stimulus set that was decorrelated on animacy dimensions taking these additional dimensions mentioned by the reviewer into account. We agree that it would be informative to test what other dimensions implicated in animacy and object recognition the five dimensions

of animacy tested in this study correlate with. We thank the reviewer for pointing towards this interesting direction for future studies, which we discussed now in the discussion section of the revised manuscript.

“Some of the dimensions tested in this study (e.g., "looking like an animal") may be correlated with the human-centred dimensions (e.g., has a face, humanness) and exploring the relationship between the five selected dimensions of animacy and the human-centred dimensions could be a focus of future studies.”

Reviewers' comments:

Reviewer #1 (Remarks to the Author):

The authors have made a valid attempt to address my previous comments. While some areas of the paper are much improved, I still think the work suffers from an attempt to draw more specific conclusions from the results than the data really support. The strengths of this work are the vast amount of data (behavior, EEG, fMRI) and the interesting use of a genetic algorithm to select stimuli. The main weakness is that the data are complicated, and do not support a clear interpretation for how such concepts may be represented in the human brain. Specifically, the selected 5 dimensions do not explain a great deal of the variance in either the EEG or fMRI data, with most of the variance in the stimulus representations remaining unexplained. This suggests that while people do use these dimensions in their categorization of these stimuli (behavior), the brain's representation of these stimuli does not map on to any of these dimensions in a straightforward manner. The revision added many supplementary figures, and I think these are useful in understanding the data better. However, I remain unsure of what the main "take home" message of the results is -- the title mentions a "disentangling" of five dimensions of animacy in brain & behavior, but given the brain & behavior do not show much agreement, it's not clear what the main conclusion is.

Specific comments:

1. The revised abstract is long and lacks focus, and it is not clear what the main conclusion of the paper is. The behavior, EEG, and fMRI data do not come together to tell a consistent story. This is fine, but the abstract makes an attempt to draw a specific conclusion about each of the five dimensions studied (and in three modalities: behavior, EEG, and fMRI-- which often do not agree with each other!) which I do not think is entirely backed up by the data. I think it would be clearer to focus on some key results placed in higher-level context, and leave more detailed speculation to the Discussion.
2. Additionally, some of the speculation in the abstract is not solidly backed up by the results. For example, it is stated that "the "being unpredictable" dimension was reflected in representations in both lower and higher-level visual cortex, possibly because unpredictable things require attention." But another, and more parsimonious explanation is that there are visual features that define the "unpredictable" category. The stimuli in Fig 4a that are highly unpredictable are also visually distinct from the other images (lighting, volcano images with distinctive coloring). Similarly, I am not sure the fMRI data supports the claim that "different brain regions sensitive to animacy may actually represent distinct dimensions". Many of the ROIs explained similar small amounts of variance for the same dimensions.
3. In terms of considering what can be concluded from large differences between brain & behavioral representations of visual stimuli, this paper by King et al 2019 (which also found little agreement between behavior & fMRI) may be useful to consider:
<https://doi.org/10.1016/j.neuroimage.2019.04.079>
4. Although I like the intention behind Figure 8 (as an attempt to visually summarize complex results), I find it a misleading characterization of the results as it suggests much more agreement between brain and behavior than the data show. It also collapses both fMRI and EEG into "brain" even though there are important differences between the fMRI and EEG data. I don't think the figure is helpful in its present form for understanding the data. The added Supp Figures are much more useful for understanding this complex data. If the authors would like to keep the figure, I'd encourage them to separate out the fMRI and EEG results and carefully consider what aspects of the data the summary emphasizes.
- 5.. p. 11, line 363: "This result means that when judging object similarity humans use all dimensions

of animacy." This claim seems a bit strong-- across all participants, there is evidence that the five dimensions all explained some unique variance in the similarity judgements (Fig 5c). But this doesn't mean that (i) all participants used all dimensions, or (ii) that the five dimensions completely captured what participants were doing in the multiarrangement task.

6. I couldn't follow some of the differences between the similar looking plots across Figure 7a, Supplementary Figures 9,10,11 and 12. Why does only Supp Fig 9 show the noise ceiling? Is Supp Fig 9 the same as Supp Fig 10 but without the general "animacy" dimension? Some x axis labels for TO2 in Supp Fig 10 are missing. Is some of the data repeated across figures?

7. I'm not sure I understand the color coding in the MDS in Supp Fig 3 / Supp Fig 14 -- do the 3 red dots for "having agency" mean that only 3 of the stimuli were rated as having agency on average?

Reviewer #2 (Remarks to the Author):

I appreciated the significant improvements in the manuscript with this revision.

There are a couple issues left to resolve:

1) for all the EEG and fMRI results (e.g., lns 438-447, lns 521-523) there is a lot of text saying that one variance was significantly different from another variance. However, the r^2 values, p values, and potential z-scores are never reported. I found it odd to have a whole results section without actually giving the means and statistical values in which statements are made.

2) The dimension of having agency seems to only have 3 stimuli with this feature. Is there a problem of low numbers here driving any results and thus are spurious?

3) Ln 426-429. I believe the MDS plots the authors are referring to are in Supplementary Fig 6? This is not noted. Also, I can appreciate MDS as being descriptive, however I did not see how "being predictable" looks any more separated than "being alive". Without it being visually obvious, or statistics able to test it, I am not sure what justification it is to make these claims.

4) In 521 - "We observed that the dimensions of animacy studied here explain variance in different brain regions to different extents, and more variance is explained by the dimensions of animacy in higher-level visual cortex" was this tested and compared? I missed where this was actually statistically tested.

5) I think it would be easy to include supplemental figure 2 as figure 5D - and would be important and more transparent to do so.

6) With regard to participant numbers for filling out objects in the animacy grid: In the caption of Fig 1, it says 11 participants were used. And then in the methods, ln 737; what does $S = 12$ refer to? Is that the number of participants? If so, why is there 12 listed, was a participant's data removed?

7) In figure 5c and 5d what does the horizontal line denote? I assume significance, but this should be noted, and in what test? A main effect? Pairwise comparisons?

8) In 533 "The results of the unique variance analysis are hard to interpret due to very low values but we included them in Supplementary Figure 15 for completeness. " what is too low? Are these values so much lower than other values you have made conclusions on?

Response to the reviewers

As outlined below, we have made revisions to address the remaining reviewers' concerns in the current manuscript. The changed text in the manuscript is in purple (revision one changes are kept in orange), addressing both reviewers' comments.

The significant improvements are as follows:

- Made the abstract shorter and more focused on the main claims of the study and edited the discussion to better articulate these (as requested by R1)
- Changed the summary Figure 8 to separate fMRI and EEG results (as requested by R1)
- Discussed some claims in more detail, adding nuances of interpretation (as requested by R1 and R2)
- Elaborated on "having agency dimension" having positive values on three stimuli (as requested by R1 and R2)
- Added more methodological details and resolved discrepancies (as requested by R2)

Please see the response to each of the reviewers for more details.

Reviewers' comments:

Reviewer #1

Referee Expertise: Neural correlates of facial recognition

The authors have made a valid attempt to address my previous comments. While some areas of the paper are much improved, I still think the work suffers from an attempt to draw more specific conclusions from the results than the data really support. The strengths of this work are the vast amount of data (behavior, EEG, fMRI) and the interesting use of a genetic algorithm to select stimuli. The main weakness is that the data are complicated, and do not support a clear interpretation for how such concepts may be represented in the human brain. Specifically, the selected 5 dimensions do not explain a great deal of the variance in either the EEG or fMRI data, with most of the variance in the stimulus representations remaining unexplained. This suggests that while people do use these dimensions in their categorization of these stimuli (behavior), the brain's representation of these stimuli does not map on to any of these dimensions in a straightforward manner. The revision added many supplementary figures, and I think these are useful in understanding the data better. However, I remain unsure of what the main "take home" message of the results is -- the title mentions a "disentangling" of five dimensions of animacy in brain & behavior, but given the brain & behavior do not show much agreement, it's not clear what the main conclusion is.

We thank the reviewer for their feedback, which has helped further improve the manuscript. We appreciate that the reviewer finds our extensive data and the approach to choosing stimuli using a genetic algorithm a strong point of the manuscript and finds that the addition of comprehensive supplementary material improved data understanding.

We agree that in the previous manuscript version, we did not adequately articulate the take-home message and did not sufficiently discuss that a portion of the variance in fMRI and EEG data remains unexplained.

As outlined below, we have made further revisions to address the reviewer's remaining concerns in the current manuscript.

Specific comments:

1. The revised abstract is long and lacks focus, and it is not clear what the main conclusion of the paper is. The behavior, EEG, and fMRI data do not come together to tell a consistent story. This is fine, but the abstract makes an attempt to draw a specific conclusion about each of the five dimensions studied (and in three modalities: behavior, EEG, and fMRI-- which often do not agree with each other!) which I do not think is entirely backed up by the data . I think it would be clearer to focus on some key results placed in higher-level context, and leave more detailed speculation to the Discussion.

We agree with the reviewer, and we have shortened the abstract and made it more focused; and discussed more detailed results and potential interpretations in the discussion.

“Distinguishing animate from inanimate things is of great behavioural importance. Despite distinct brain and behavioural responses to animate and inanimate things, it remains unclear which object properties drive these responses. Here, we investigate the importance of five object property dimensions related to animacy (“being alive”, “looking like an animal”, “having agency”, “having mobility”, and “being unpredictable”) in brain (fMRI, EEG) and behaviour (property and similarity judgements) of 19 participants. We use a stimulus set of 128 images, optimized by a genetic algorithm to disentangle the five chosen dimensions. We find that these dimensions explained much variance in the similarity judgments. Each dimension also explained significant variance in the brain representations, except, surprisingly, “being alive”. Our results suggest that different brain regions sensitive to animacy may represent distinct dimensions, either as accessible perceptual stepping stones toward detecting whether something is alive or because they are of behavioural importance by themselves.”

2. Additionally, some of the speculation in the abstract is not solidly backed up by the results. For example, it is stated that "the “being unpredictable” dimension was reflected in representations in both lower and higher-level visual cortex, possibly because unpredictable things require attention." But another, and more parsimonious explanation is that there are visual features that define the "unpredictable" category. The stimuli in Fig 4a that are highly unpredictable are also visually distinct from the other images (lighting, volcano images with distinctive coloring). Similarly, I am not sure the fMRI data supports the claim that "different brain regions sensitive to animacy may actually represent distinct dimensions". Many of the ROIs explained similar small amounts of variance for the same dimensions.

We thank the reviewer for the opportunity to clarify. We have controlled for the low-level stimuli confounds in the revised manuscript by including the early layer of a deep neural

network as regressors in the unique variance analysis and showed that the animacy dimensions (including “the being unpredictable” dimension) explained unique variance in the data (Supplementary Figures 2, 7, 16). However, we agree with the reviewer that the explanation of the “being unpredictable” dimension requiring attention is speculative. Therefore, we removed this sentence from the abstract (see the response to point one).

While we agree that the amount of variance explained by the dimensions was not high, these dimensions did explain variance in the brain responses. Differences existed (some dimensions did not explain variance in early visual cortex, but they did in higher-level visual cortex). Following the reviewer’s suggestion, we now included a sentence about the amount of variance explained in brain representations being smaller than in behaviour in the revised abstract.

“Brain representations were also explained by most dimensions (surprisingly not “being alive”), however, to a lesser extent than behaviour.”

3. In terms of considering what can be concluded from large differences between brain & behavioral representations of visual stimuli, this paper by King et al 2019 (which also found little agreement between behavior & fMRI) may be useful to consider:

<https://doi.org/10.1016/j.neuroimage.2019.04.079>

We thank the reviewer for bringing our attention to this important article. We have discussed and cited the article in the revised manuscript.

“Little agreement between behavioural similarity judgements and 7T fMRI responses has also been found in King et al., 2019.”

4. Although I like the intention behind Figure 8 (as an attempt to visually summarize complex results), I find it a misleading characterization of the results as it suggests much more agreement between brain and behavior than the data show. It also collapses both fMRI and EEG into “brain” even though there are important differences between the fMRI and EEG data. I don’t think the figure is helpful in its present form for understanding the data. The added Supp Figures are much more useful for understanding this complex data. If the authors would like to keep the figure, I’d encourage them to separate out the fMRI and EEG results and carefully consider what aspects of the data the summary emphasizes.

We agree that Figure 8 may be confusing given the fMRI and EEG results merge. Therefore, we have separated the fMRI and EEG results in the revised Figure 8 (see below). Following the reviewer’s suggestion, we also added a sentence and visually indicated on the figure that the five tested dimensions explained less explainable variance in behaviour than in brain representations.

“We find that the five tested dimensions of animacy captured behaviour very well. Brain representations were also explained by most dimensions (surprisingly not “being alive”), however, to a lesser extent than behaviour.”

5. p. 11, line 363: "This result means that when judging object similarity humans use all dimensions of animacy." This claim seems a bit strong-- across all participants, there is evidence that the five dimensions all explained some unique variance in the similarity judgements (Fig 5c). But this doesn't mean that (i) all participants used all dimensions, or (ii) that the five dimensions completely captured what participants were doing in the multiarrangement task.

(i) We agree with the reviewer that we cannot be sure whether all participants used all dimensions. We use Representational Similarity Analysis to draw conclusions based on data across participants, and we do not explore individual differences in detail. However, looking at the error bars in the Representational Similarity Analysis (Figure 5c), the error bars are small, which suggests that all participants used these dimensions to a certain degree. (ii) We agree that other dimensions are needed to fully capture the variance in the similarity judgements task, and we further discussed this point.

"None of the dimensions fully explained the similarity judgements data but the "having agency" dimension was close to explaining the total explainable variance given the noise in the data. As a portion of the variance remained unexplained, other dimensions beyond the ones explored here are likely needed to capture the data fully. Overall, when judging object similarity, humans use all dimensions of animacy tested here."

6. I couldn't follow some of the differences between the similar looking plots across Figure 7a, Supplementary Figures 9,10,11 and 12. Why does only Supp Fig 9 show the noise ceiling? Is Supp Fig 9 the same as Supp Fig 10 but without the general "animacy" dimension? Some x axis labels for TO2 in Supp Fig 10 are missing. Is some of the data repeated across figures?

We included these different analysis variants following the suggestion from reviewer 2. Displaying the noise ceiling was requested by reviewer 2, and therefore, we included this figure in the supplementary materials. As the noise ceiling is high, which makes it hard to

see the correlation values, and consequently, we did not include it in the other plots for easier readability. The differences between the similar-looking figures are as follows: Figure 7a, Supplementary Figures 9,10,11 and 12. We do indicate what each of these figures represents in the text.

“We first evaluated the contribution of each animacy dimension in ROIs across the ventral: visual area 1 (V1v), ventral occipital cortex 2 (VO2), parahippocampal cortex 2 (PHC2) and dorsal: visual area 1 (V1d), lateral occipital cortex 2 (LO2), TO2 visual streams (Figure 7a, Supplementary Figure 9 - with displayed noise ceiling).”

“After examining the contribution of general “being animate” ratings to the fMRI ROI representations, we did not see that “being animate” explained a significant amount of variance in fMRI data (Supplementary Figure 10). For completeness, we performed the same analysis for all regions in ventral (V1v, V2v, hV4, VO1, VO2, PHC1, PHC2) and dorsal (V1d, V2d, V3d, V3a, V3b, LO1, LO2, TO1, TO2) visual streams. We observed that more dimensions of animacy studied here explain variance in higher-level visual cortex than in early visual cortex (Supplementary Figures 11 and 12).”

We thank the reviewer for pointing out the missing x-axis labels for TO2 in Supplementary Figure 10; we have added them in the revised manuscript.

7. I'm not sure I understand the color coding in the MDS in Supp Fig 3 / Supp Fig 14 -- do the 3 red dots for "having agency" mean that only 3 of the stimuli were rated as having agency on average?

Yes. We thank the reviewer for stressing that a discussion is needed for “having agency” having positive values for three stimuli (images of two adult humans and a human fetus in late stages of pregnancy). This dimension explained a significant amount of variance in the general animacy ratings, similarity judgements, EEG, and fMRI data, so it does not seem this number of stimuli was a problem. We have discussed this in the manuscript text:

“Despite “having agency” dimension having positive values for three stimuli (images of two adult humans and a human fetus in the late stages of pregnancy, Supplementary Figure 1), this dimension of animacy explained a lot of variance in the similarity judgements (Figure 5c).”

Reviewer #2

Referee expertise: Neural correlates of visual perception

I appreciated the significant improvements in the manuscript with this revision.

We appreciate that the reviewer finds the manuscript markedly improved after revision. As outlined below, we have made further revisions to address the remaining reviewer's concerns in the current manuscript version.

There are a couple of issues left to resolve:

1) for all the EEG and fMRI results (e.g., lns 438-447, lns 521-523) there is a lot of text saying that one variance was significantly different from another variance. However, the r^2 values, p values, and potential z-scores are never reported. I found it odd to have a whole results section without actually giving the means and statistical values in which statements are made.

The reviewer refers to lines 438-447:

“Despite differences in the exact timing of when dimensions of animacy explained the variance, a very clear pattern that one dimension explains representations earlier than the other was not observed. However, 'being unpredictable' explained significantly more variance than most dimensions in early time points: specifically more than “looking like an animal” (89-130 ms), “having mobility” (89-113 ms), and “having agency” (79-126 ms). While “looking like an animal” explained more variance than most other dimensions in later time points: more than “being alive” (209-302 ms), “having agency” (230-266 ms), and “being unpredictable” (146-184 ms). Finally, “having agency” explained more variance than most of the dimensions even later in time: more than “being alive” (268-301 ms), “having mobility” (261-289 ms) and “being unpredictable” (293-315 ms).”

Because of the nature of the EEG data (temporal), it is not possible to report a single r-value or p-value and reporting a range of them (at each ms) seems impractical. The r-values can be read from Figure 5. The statistical calculations are described in the methods section and Figure 5 legend but following the reviewer's comment we added them to the main manuscript text.

“Once we knew that we could decode our stimuli, we asked how much variance each animacy dimension explained in EEG recordings. Do any of the dimensions explain any variance at all? To answer this question, we correlated each animacy dimension with EEG representations at every time point in every participant using RSA. First, we determined whether each of the animacy dimension RDMs was significantly related to the EEG data RDMs at every timepoint using a participant-as-random-effect analysis (one-sided Wilcoxon signed-rank test). We subsequently tested for differences in animacy dimension performance between each pair of dimensions of animacy at each timepoint using a participant-as-random-effect analysis (two-sided Wilcoxon signed-rank test). We accounted for multiple comparisons for each analysis by controlling the FDR at 0.05.”

Lines 521-523 have been removed following the reviewer's comment 4 (please see our response there).

2) The dimension of having agency seems to only have 3 stimuli with this feature. Is there a problem of low numbers here driving any results and thus are spurious?

We thank the reviewer for stressing that a discussion is needed for “having agency” having positive values for three stimuli (images of two adult humans and a human fetus in late stages of pregnancy). This dimension explained a significant amount of variance in the general animacy ratings, similarity judgements, EEG, and fMRI data, so it does not seem this number of stimuli was a problem. We discuss this in the current version of the manuscript:

“Despite “having agency” dimension having positive values for three stimuli (images of two adult humans and a human fetus in the late stages of pregnancy, Supplementary Figure 1), this dimension of animacy explained significant variance in the similarity judgements (Figure 5c).”

3) Ln 426-429. I believe the MDS plots the authors are referring to are in Supplementary Fig 6? This is not noted. Also, I can appreciate MDS as being descriptive, however I did not see how "being predictable" looks any more separated than "being alive". Without it being visually obvious, or statistics able to test it, I am not sure what justification it is to make these claims.

We thank the reviewer for pointing out that the reference to Supplementary Figure 6 is missing. We have added it. We removed the sentence discussing the "being predictable" dimension looking more separated than "being alive". We agree that the data did not strongly back up this conclusion.

“We have also displayed MDS plots colour-coded according to binary animacy dimensions for visualization purposes (Supplementary Figure 6).”

4) In 521 - "We observed that the dimensions of animacy studied here explain variance in different brain regions to different extents, and more variance is explained by the dimensions of animacy in higher-level visual cortex" was this tested and compared? I missed where this was actually statistically tested.

We agree that this statement was not backed up by statistical tests; we rephrased the conclusion to reflect the results more accurately”

“We observed that more dimensions of animacy studied here explain variance in higher-level visual cortex than in early visual cortex (Supplementary Figures 11 and 12).”

5) I think it would be easy to include supplemental figure 2 as figure 5D - and would be important and more transparent to do so.

We agree with the reviewer that the analysis where we include an early layer of a deep neural network in the unique variance analysis is an important control. Therefore, we have included this control analysis for all datasets in the manuscript's revised version. However,

as this control analysis did not change the pattern of the results for the unique variance analysis in similarity judgements, we decided to keep it in the supplementary materials.

6) With regard to participant numbers for filling out objects in the animacy grid: In the caption of Fig 1, it says 11 participants were used. And then in the methods, In 737; what does S = 12 refer to? Is that the number of participants? If so, why is there 12 listed, was a participant's data removed?

We thank the reviewer for spotting this inconsistency. Eleven participants filled in the grid. We corrected this information in the text.

“We created a grid with all possible combinations of dimensions of animacy ($2^5 = 32$). We asked participants ($S = 11$, mean age = 33, 6 females) to write down object category names (e.g., “humanoid robot”) for each combination in the grid to obtain a list of object categories (Figure 1a Step 1).”

7) In figure 5c and 5d what does the horizontal line denote? I assume significance, but this should be noted, and in what test? A main effect? Pairwise comparisons?

We previously stated “using the same conventions as in Figure 4b” where we included the details of what the horizontal line depicts. We pasted the information below:

“Bars show the correlation between the animacy ratings RDMs and each animacy dimension RDM. A significant correlation is indicated by an asterisk (one-sided Wilcoxon signed-rank test, $p < 0.05$ corrected). Error bars show the standard error of the mean based on single-participant correlations, i.e., correlations between the single-participant animacy ratings RDMs and animacy dimension RDM. The grey bar represents the noise ceiling, which indicates the expected performance of the true model given the noise in the data. Horizontal lines show pairwise differences between model performance ($p < 0.05$, FDR corrected across all comparisons).”

8) In 533 "The results of the unique variance analysis are hard to interpret due to very low values but we included them in Supplementary Figure 15 for completeness. " what is too low? Are these values so much lower than other values you have made conclusions on?

We agree with the reviewer that what “too low” means is arbitrary. Therefore, we have removed this sentence and described the results of the unique variance analysis as detailed below:

“The results of the unique variance analysis are included in Supplementary Figure 15. “Being unpredictable”, “having mobility”, and “having agency” dimensions explained unique variance in early visual cortex. Except “being alive” dimension, four dimensions of animacy explained unique variance in higher-level visual areas.”

REVIEWERS' COMMENTS:

Reviewer #1 (Remarks to the Author):

1. There is a discrepancy between the manuscript and the response to reviewers document in response to my previous point 2:

(from response letter)

"Brain representations were also explained by most dimensions (surprisingly not "being alive"), however, to a lesser extent than behavior."

(from manuscript)

"Each dimension also explained significant variance in the brain representations, except, surprisingly, "being alive".

The response letter states that "Following the reviewer's suggestion, we now included a sentence about the amount of variance explained in brain representations being smaller than in behavior in the revised abstract." but this point is NOT actually made in the revised abstract.

The fact that much of the variance in brain responses to these stimuli remains unexplained is an important take-away from the data, and this should be made clear in the manuscript.

2. Unfortunately the revised Figure 8 still provides a misleading summary of the data, and I recommend removing it. Of the 5 dimensions, only 'looking like an animal' explained unique variance in the EEG data. In the fMRI data, 3 dimensions explained unique variance in early visual cortex and 4 dimensions in higher visual cortex. Figure 8 is not consistent with this important pattern of results and gives a completely different impression of the data (e.g. EEG data has a tick for 4 dimensions, even though only 1 explained unique variance?)-- it overemphasizes agreement between the imaging modalities and does not summarize the critical features of the results (which are in the details in this case).

Reviewer #2 (Remarks to the Author):

The authors have satisfactorily addressed my comments.